# Object Reprojection Error (ORE): Camera pose benchmarks from lightweight tracking annotations

**Xingyu Chen**[1,*], **Weiyao Wang**[1,*], **Hao Tang**[1], **Matt Feiszli**[1]
FAIR, Meta[1], Equal Technical Contribution[*]
{xingyuchen, weiyaowang, haotang, mdf}@meta.com

## Abstract

3D spatial understanding is highly valuable in the context of semantic modeling of environments, agents, and their relationships. Semantic modeling approaches employed on monocular video often ingest outputs from off-the-shelf SLAM/SfM pipelines, which are anecdotally observed to perform poorly or fail completely on some fraction of the videos of interest. These target videos may vary widely in complexity of scenes, activities, camera trajectory, etc. Unfortunately, such semantically-rich video data often comes with no ground-truth 3D information, and in practice it is prohibitively costly or impossible to obtain ground truth reconstructions or camera pose post-hoc.

This paper proposes a novel evaluation protocol, Object Reprojection Error (ORE) to benchmark camera trajectories; ORE computes reprojection error for static objects within the video and requires only lightweight object tracklet annotations. These annotations are easy to gather on new or existing video, enabling ORE to be calculated on essentially arbitrary datasets. We show that ORE maintains high rank correlation with standard metrics based on ground truth. Leveraging ORE, we source videos and annotations from Ego4D-EgoTracks, resulting in EgoStatic, a large-scale diverse dataset for evaluating camera trajectories in-the-wild.

## 1 Introduction

Spatial understanding is a fundamental tool in human perception [49]. In computer vision, spatial understanding tasks often come in the form of geometric reconstruction and localization [62, 76, 74], typically seeking precise camera localization and 3D scene maps. The focus is on metric structure, not relations and objects. On the other hand, neurophysiology suggests human memory is anchored in object-centric or allocentric coordinates [10, 88, 26, 32]. This interpretation suggests a different granularity of spatial modeling and is relevant to applications in human-in-the-loop settings, like augmented reality. For example, equipped with moderately-accurate object-level spatial understanding, an AR assistant can give guidance to a human performing a task, describe object locations and spatial relations, and infer user intention.

Indeed, localizing an agent relative to objects and environment has become foundational in many recent approaches to semantic understanding, e.g. spatial semantic mapping [14, 34, 64, 77, 59, 79], place categorization [83], 3D scene graph construction [4, 69, 70], spatial episodic memory [5, 45, 37, 6, 48, 65, 101], visual query [42, 43, 57, 98, 56], planning and guiding visual navigation (e.g. for AR) [17, 2, 45, 15, 16], representation learning in egocentric vision [28, 50, 90, 60, 85], action anticipation [44, 68, 66] and active object segmentation [89]. These methods use spatial information at a higher semantic level; exact locations of pixels and cameras are less critical. They focus on the spatial semantics and relations of objects and cameras. In practice however, these approaches typically rely on careful 3D geometry, in the form of depth, camera trajectories and volumes, but often discard much of the accuracy in favor of looser notions of proximity (e.g. [4, 14, 34]).

37th Conference on Neural Information Processing Systems (NeurIPS 2023) Track on Datasets and Benchmarks.

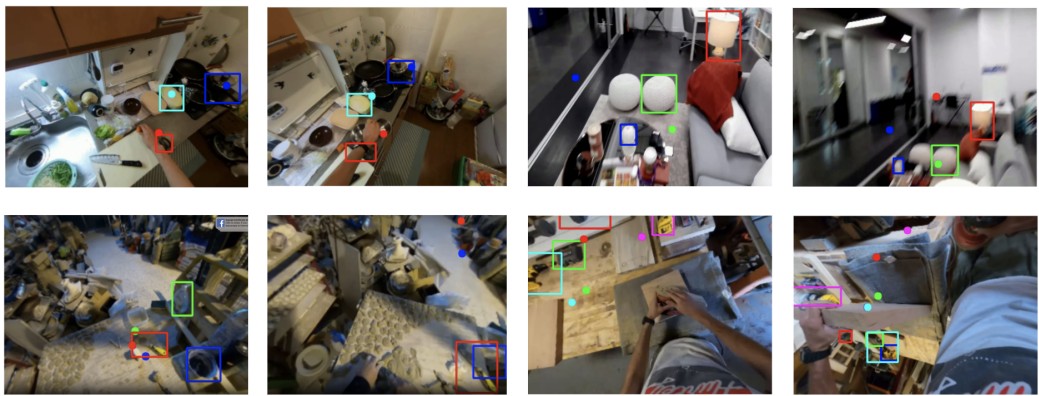

Figure 1: **EgoStatic Benchmark Visualization** Top row: COLMAP, bottom row: DROID-SLAM. Colored bounding boxes are ground truth tracklet annotations; dots are reprojected points tracked by estimated camera trajectory. Ideally, the points should fall in the bounding boxes.

Unfortunately, many common datasets for semantic tasks have no 3D ground truth of any kind, and it is expensive or impossible to obtain it post-hoc. How well can the underlying systems localize agents, objects and their relations in these domains? Studies on real-world ego-centric videos have often been limited to synthetic data [96] or on scan-style/walking videos [21, 72] vs. real-world videos with activities (e.g. cooking, DIY, sports) with rich semantics. While prior work [43, 57, 89, 64, 104] notes SLAM/ SfM failures, e.g. on head-mounted videos, rarely is this quantified (nor is it typically possible). In other words, we have high-value problems whose solutions currently depend on systems which cannot be benchmarked in the domains of interest.

**Contributions.** This work proposes a novel, object-centric metric for camera trajectory quality on essentially arbitrary video (e.g. videos in Fig. 1). As a measure of environmental understanding, the resulting benchmark uses a type of reprojection error which is a generalization of geometric reprojection error; in this way it is similar to traditional SfM / SLAM benchmarks. However, using a sparse set of semantic landmarks, specifically static objects whose identities are unknown to the method under evaluation, keeps the benchmark focused on high-level percepts in the vicinity of the camera, vs. global maps and reconstruction accuracy.

Object Reprojection Error (ORE) relaxes the need for accurate groundtruth camera trajectories. Given an arbitrary video, one identifies a few "suitable" object candidates and annotates 2D bounding boxes across the frames where the object remains unmoved (Fig. 1). Despite no groundtruth camera trajectory, ORE's rank statistics agree well with standard GT-based metrics, such as Absolute Trajectory Error (ATE) and Relative Pose Error (RPE), when properly-selected tracklets are used.

Equipped with this schema, we source a wide variety of ego-centric videos from Ego4D, a large-scale egocentric video collected in-the-wild. We leveraged the long-term tracking annotations from EgoTracks [86] on Ego4D. Our contributions include:

1. We carefully design a new evaluation protocol for camera trajectory estimation which only requires static object tracklet annotation and no camera trajectory groundtruth: ORE.
2. We benchmark 7 SLAM, Visual Odometry (VO) and Structure-from-Motion (SfM) methods on Scannet test set [21] to compare ORE with standard metrics. Rank correlation shows high agreement between ORE and standard metrics.
3. We extend and adapt Ego4D-EgoTracks [86] benchmark into a first large-scale egocentric camera trajectory benchmark. The resulting benchmark is shown to be quite challenging.
4. Finally, ORE is a useful tool: it is sensitive enough to inform hyperparameter selection and method design. The experiments reveal potential directions for future work.

## 2 Related Work

### 2.1 SLAM, SfM and Visual Re-localization

SLAM (Simultaneous Localization And Mapping) and SfM (Structure-from-Motion) aim to jointly infer a map and localize an agent within it. SLAM systems often use multiple sensors including RGB [62], depth [63] or inertial sensor [67, 92]. They can be categorized as direct [31, 30, 106] or indirect methods [23, 62, 63, 12, 53]. Direct methods utilizes pixel intensities directly and estimate 3D geometry by minimizing the photometric error; indirect methods extract intermediate representations like point features [62], then optimize camera poses and 3D point clouds via reprojection error. COLMAP [74] is a widely-adopted, general-purpose SfM pipeline, often the default method in higher-level tasks [89, 43, 57, 56]. **Deep learning** has recently been integrated into SLAM system, producing reliable and view-invariant features [82, 71, 39] and direct pairwise pose predictions in visual odometry [94, 99, 55, 54, 40]. The end-to-end SLAM system, DROID-SLAM [87] proposes an iterative update operator for camera poses and dense depth, using a Dense Bundle Adjustment layer. ParticleSfM [104] leverages deep learning to classify static vs dynamic point trajectories and combines with classical SfM [84]. We evaluate a representative set of these methods in what follows. Visual Re-localization [13, 78, 73] aims to recover the 6DoF camera poses of query RGB images from a known environment represented by feature embeddings [51, 3], 3D point cloud [8, 58], scene coordinates [97, 105, 9] or Neural Radiance Field (NeRF) [19, 18]. However, these relocalization methods require pose ground truth to train for unknown scenario [19, 51] and mostly focus on single-image pose estimation [51, 105], thus are not the best evaluation candidates for EgoStatic benchmark.

### 2.2 Datasets and Benchmarks

Most existing camera trajectory benchmarks [36, 21, 11] require accurate ground truth camera trajectories to compare with. This often relies on additional sensors other than RGB cameras: rotating 3D laser scanner and combined GPS/IMU inertial navigation system (KITTI [36]), RGB-D and IMU data (ScanNet [21]), LiDAR depth/points, IMU and radio data (LaMAR [72]). Bulky hardware can limit such datasets' domains to driving [36, 20], drones [11, 81], indoor and outdoor scans [72, 21] and synthetic data [96]. Other works [27, 80] rely on SfM to extract pose on crowd-sourced images of the same scene. We compare these benchmarks with our proposed Egostatic in Table 1: the diversity and scale of EgoStatic serve as a strong complement to existing benchmarks.

Table 1: EgoStatic provides the largest-scale evaluation, with diverse scenes and complex actions.

| | Seq. | Total Size | Scenario |
|---|---|---|---|
| KITTI [36] | 22 | 13k frames | Driving |
| EuRoC MAV [11] | 11 | 23 minutes | Indoor drone |
| TUM RGB-D [81] | 15 | 27 minutes | Indoor drone |
| DISCOMAN [52] | 200 | 5k frames | Indoor |
| TartanAir [96] | 1037 | 1M frames | Synthetic |
| LaMAR [72] | 300 | 100 hours | Walking in 3 sites |
| ScanNet [21] | 1500 | 2.5M frames | Indoor room scan |
| Ours (EgoStatic) | **5708** | **600 hours / 9M frames** | **137 Everyday Life scenarios** |

### 2.3 SLAM Evaluation Metrics

Most SLAM [62] or SfM [33] systems produce per-frame camera pose. Various metrics have been proposed: Absolute Trajectory Error (ATE) [103] performs a global positional alignment [91] between estimated and ground truth poses, then calculates RMSE (root mean square error). Relative Pose Error (RPE) [103] computes local errors across sub-trajectories in a temporal sliding window. Recently, alternative SLAM metrics have also been proposed. ODE(Overlap Displacement Error) [61] moves away from pose error, instead evaluating map inconsistency introduced by the pose errors. Bodin [7] evaluates reconstruction error after ICP alignment. Our proposed metric ORE is related to the classic metric reprojection error [46], but we focus specifically on object-centric reprojection.

# 3 EgoStatic Benchmark

In this section, we introduce EgoStatic, a benchmark for evaluating camera trajectory relative to objects. We first introduce an evaluation pipeline leveraging static objects' tracklets to compute Object Reprojection Error (ORE). This pipeline does not depend on groundtruth camera poses or sensors beyond RGB camera. The pipeline can be applied to any video, even after collection, enabling evaluation on existing large-scale egocentric data. We next provide geometric intuitions and key factors in dataset construction; these tighten the connection between ORE and classic metrics. Empirical studies (sec 4) validate the correlation. Finally, we detail the additional annotations and adopt this pipeline to Ego4D-EgoTracks video. This results in EgoStatic, with 5708 video sequences, over 9 million frames and 22k static object tracklets. To our knowledge, this is by far the largest real-world evaluation benchmark of its kind.

## 3.1 Existing camera trajectory metrics

We revisit two frequently used metrics: Absolute Trajectory Error (ATE), and Relative Pose Error (RPE) [103]. Their goal is to measure the quality of a series of $N$ estimated camera poses $\hat{\mathbf{X}} = \{\hat{\mathbf{R}}_i, \hat{\mathbf{t}}_i\}_{i=0}^{N-1}$ (rotation $\hat{R}_i \in SO(3)$, translation $\hat{t}_i \in \mathbb{R}^3$) vs. ground truth $\mathbf{X}^{gt} = \{\mathbf{R}_i^{gt}, \mathbf{t}_i^{gt}\}_{i=0}^{N-1}$.

**ATE** is a global measure of squared error between estimated trajectory $\hat{\mathbf{X}}$ and ground truth $\mathbf{X}^{gt}$. Due to coordinate system ambiguity[91], an optimal alignment $g^*(\cdot) = \{s, R, t\}$ is first applied:

$$g^*(\cdot) = \underset{g=\{s,R,t\}}{\arg\min} \sum_{i=0}^{N-1} ||\mathbf{t}_i^{gt} - s\mathbf{R}\hat{\mathbf{t}}_i - t|| \tag{1}$$

ATE for rotation and translation can be calculated as the root mean square error (RMSE):

$$ATE_{trans} = \left( \frac{1}{n} \sum_{i=0}^{N-1} ||\mathbf{t}_i - g^*(\hat{\mathbf{t}}_\mathbf{i})||^2 \right)^{\frac{1}{2}} ; ATE_{rot} = \left( \frac{1}{n} \sum_{i=0}^{N-1} ||\angle(\mathbf{R}_i(g^*(\hat{\mathbf{R}}_\mathbf{i})))^T||^2 \right)^{\frac{1}{2}} \tag{2}$$

where $\angle(\cdot)$ means the angle difference between two rotation matrices.

**RPE** is a local consistency metric calculated by the trajectory over a fixed interval $\Delta$. Given a series of ground truth camera poses $P_i \in SE(3)$ and estimated poses $Q_i \in SE(3)$, both within the local window, the relative pose error matrix can be defined as:

$$E_i^\Delta = (Q_i^{-1}Q_{i+\Delta}^{-1})(P_i^{-1}P_{i+\Delta}) \tag{3}$$

With $E_i^\Delta = [R_i^\Delta | t_i^\Delta]$, the rotation and translation component of RPE can be computed as:

$$RPE_{rot} = \left( \frac{1}{m} \sum_{i=0}^{m-1} \arccos\left( \frac{tr(R_i^\Delta) - 1}{2} \right) \right) ; RPE_{trans} = \left( \frac{1}{m} \sum_{i=0}^{m-1} ||t_i^\Delta||^2 \right)^{\frac{1}{2}} \tag{4}$$

## 3.2 Evaluation methodology

Given $N$ estimated camera poses $\hat{X}$ and $M$ ground truth static object tracklets (e.g. bounding boxes) $\{B_i\}_{i=0}^{M-1}$, we aim to quantify the quality of $\hat{X}$, *without* ground truth pose $X^{gt}$. We summarize our entire procedure in Algorithm 1 and Figure 2. The pipeline contains two steps (Fig. 2): (a) Lift Object to 3D and (b) Re-Projecting Object down to 2D.

**(a) Lift Object to 3D** For each tracklet $B_i$, we select an arbitrary occurrence of object $i$ at frame $j$, resulting a segmentation mask or a bounding box $b_{i,j}$. We now need to identify a point on the object. If masks are available we pick a central point in the mask (ScanNet), otherwise we use point $\hat{p}$ at the box center (EgoTracks). If the depth $\hat{d}$ of the point in this frame were known,then given camera intrinsics $K$, we can unproject $\hat{p}$ to its world location

$$\hat{p}_{3d} = [R, T]^{-1}K^{-1}\hat{p}, \text{ where } \hat{p} = (x_0, y_0, 1, 1/\hat{d})$$

**(b) Project to 2D** With 3D location $\hat{p}_{3d}$ of a point on the object, we can project to 2D for other frames to generate a point tracklet $\{\hat{p}_k\}_{k=0}^{N-1}$ by applying the camera projection formula on estimated

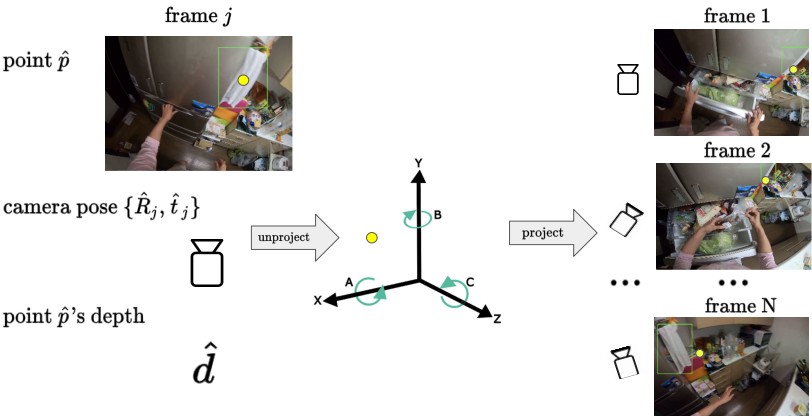

Figure 2: **Point Tracklet Generation** Point tracklets are obtained by lifting and reprojecting a sampled object point $\hat{p}$, given camera trajectory $\hat{X} = \{\hat{R}_i, \hat{t}_i\}_{i=0}^{N-1}$, and point depth $\hat{d}$

camara poses $\hat{X}_k = \{\hat{R}_k, \hat{t}_k\}$, $\hat{p}_k = K\hat{X}_k\hat{p}_{3d}$. Since the object is static, the reprojected point will fall inside the bounding box or the segmentation mask if the estimated pose is good. We define Object Reprojection Error (ORE) as the distance between the reprojected point location $\{\hat{p}_k\}_{k=0}^{N-1}$ and the annotated object 2D location $\{b_{i,k}\}_{k=0}^{N-1}$.

$$\mathbf{ORE}(\hat{p}_k, b_{i,k}) = \begin{cases} 0, & \text{if } \hat{p}_k \in b_{i,k} \\ ||\hat{p}_k - b_{i,k}||, & \text{if } \hat{p}_k \notin b_{i,k} \end{cases} \tag{5}$$

We use $l1$-norm and normalize by the image size. **ORE** is averaged for all tracklets in a sequence.

**Depth Optimization.** In step (a), we assumed depth is known at frame $j$ for the selected point. In practice, ground truth depth is not generally available, and any attempt to infer it could introduce model bias or systematic error, skewing results. Instead, we propose an optimization: we select the depth value $d^*$ that minimizes **ORE** for each camera trajectory. This is similar to the global alignment in the ATE metric which minimizes RMSE, yet with fewer degrees of freedom to optimize. We tried multiple optimizers [93, 29], yet found simple grid-search sufficient to find a good value.

---

**Algorithm 1** Calculate Object Re-Projection (ORE) Error

---

> **function ORE-D**$(\{B_i\}_{i=0}^{M-1}, \hat{X} = \{\hat{R}_i, \hat{t}_i\}_{i=0}^{N-1}, \text{d})$
>> **for** tracklet $B_i \in \{B_i\}_{i=0}^{M-1}$ **do**
>>> Sample point $\hat{p}_j = (x_0, y_0)$ from $b_{i,j} \in B_i$ in arbitrary frame $j$
>>> $\hat{p}_{3d} = \hat{X}_j^{-1}K^{-1}(x_0, y_0, 1, 1/d)^T$
>>> **ORE** $= \{\}$
>>> **for** frame $k$ in video **do**
>>>> $\hat{p}_k = K\hat{X}_k\hat{p}_{3d}$
>>>> **ORE** append **ORE**$(\hat{p}_i, b_{i,k})$
>>> **end for**
>> **end for**
>> **return** $\frac{1}{M}\sum_i \frac{1}{K}\sum_k \mathbf{ORE}(\hat{p}_i, b_{i,k})$
> **end function**
> **function ORE**$(\{B_i\}_{i=0}^{M-1}, \hat{X} = \{\hat{R}_i, \hat{t}_i\}_{i=0}^{N-1})$
>> **ORE** $= \underset{d}{\text{minimize}} \mathbf{ORE\text{-}D}(\{B_i\}_{i=0}^{M-1}, \hat{X} = \{\hat{R}_i, \hat{t}_i\}_{i=0}^{N-1}, d)$
> **end function**

---

## 3.3 Constructing a benchmark

When constructing an ORE benchmark, the key consideration is the choice of tracklets and camera trajectories. Here we discuss criteria for selecting good videos.

What configurations of objects will be most effective at capturing errors in camera trajectory? Ideally, a good configuration would be maximally sensitive to errors, i.e. it will reliably produce signal which grows with error magnitude. One way to identify such configurations would be, for example, to compute a probability distribution on possible errors (e.g. a distribution on SE(3) centered about the identity), and identify configurations of points which maximize the expected reprojection error under this distribution. However, we are ultimately limited by the tracklets we have available, so this feels like overkill; relatively simple intuition is sufficient to guide our choice of videos.

First, observe that a configuration of points which lies on a single camera ray carries no information about either translation error in the ray's direction or roll error about the ray; such degenerate configurations should be avoided. Further, we see that sensitivity to roll error about the optical axis, as well as pitch/yaw error, both increase as points move away from the optical center. Pitch/yaw error sensitivity is maximized when points are aligned with the rotational vector field. Putting this all together suggests a set of objects, well-distributed in angle and located relatively far from the optical center, will be sensitive to the widest possible set of rotations and will maximize overall response to error. Choosing objects with smaller bounding boxes will further reduce the nullspace of ORE and should also improve sensitivity. In what follows we employ heuristics to choose appropriate videos and study these factors empirically.

### 3.4 EgoStatic benchmark

To evaluate camera trajectories in real-world environment with complex activities, we source videos from Ego4D dataset [43] and object tracking annotations from EgoTracks dataset [86]. Since EgoTracks covers both active and static objects, we annotate additional attribute labels on each object occurrence of EgoTracks, identifying the static object tracklets. This results in around 22,000 static object tracklets from 5708 6-minutes video sequences ($\sim$600 hours).

## 4 Experiments

Here, we benchmark a representative set of methods using ORE. We divide our experiments into two sets. First, on ScanNet (which contains GT camera pose) we (i) show ORE is strongly correlated to well-established metrics and (ii) examine our design choices, motivated by geometric intuitions in section 3.3. Next, we turn to our proposed EgoStatic benchmark. We show that EgoStatic is a challenging benchmark, and second that ORE is a strong tool: it is sensitive enough to enable hyper-parameter selection and model design choices.

### 4.1 Baseline methods and implementations

We select 7 methods from SLAM, VO and SfM as baselines. Unless otherwise specified, we use the default setup from the original work. For SLAM, we use both geometry-based approaches such as ORB-SLAM2 [63] and ORB-SLAM3 [12] and SOTA learning-based approach DROID-SLAM [87]. For visual odometry, we include self-supervised MonoDepth2[1] [41] and synthetic-data-supervised TartanVO [95]. For SfM, we include both geometry-based approach COLMAP [74] and hybrid method ParticleSfM [104] that combines learned correspondence and geometry mapping.

**Tuning COLMAP.** As observed in prior works [104, 89, 57], default COLMAP may fail entirely on ScanNet or egocentric video, or converge slowly with poor quality output. Following [89], we replace the exhaustive matching procedure with vocabulary tree matching procedure.

### 4.2 Experiments on ScanNet

**ScanNet** [21] is an RGB-D video dataset containing more than 1500 3D scans, each with 3D camera pose annotations and point-clouds. We pick high confidence instance mask from Mask-RCNN [47], and construct object tracklets by back-projecting masks to the point cloud. We benchmark methods on 20 test sequences on ATE, RPE and ORE (Tab. 2; see suppl. for per-scene results).

---

[1]No part of Meta's research involved any use of MonoDepth2; The data pertaining to MonoDepth2 is included solely for comparison purposes.

Table 2: ORE ranks methods consistently with $ATE_{trans}$ and $ATE_{rot}$. Average ORE, $ATE_{trans}$ and $ATE_{rot}$ performance of 7 methods on ScanNet dataset, $ATE_{trans}$, $ATE_{rot}$, and ORE are measured in meters, radians and normalized unit distance between 0 and 1 respectively.

|  | DROID[87] | COLMAP[33] | ORB2 [63] | ORB3 [12] | Particle [104] | TartanVO [95] | Monodepth2 [41] |
|---|---|---|---|---|---|---|---|
| ORE | 0.029 | 0.035 | 0.102 | 0.109 | 0.130 | 0.251 | 0.294 |
| $ATE_{trans}$ (m) | 0.147 | 0.331 | 0.462 | 0.533 | 0.484 | 0.451 | 0.790 |
| $ATE_{rot}$ (rad) | 0.135 | 0.448 | 0.896 | 1.065 | 0.629 | 2.096 | 1.861 |

Table 3: **ORE has strong rank correlation with $ATE_{trans}$ and $ATE_{rot}$ on ScanNet.** Spearman Coef. (upper triangle) and Kendall Coef. (lower triangle) between ORE and standard metrics.

| Spearman / Kendall | ORE | $ATE_{trans}$ | $ATE_{rot}$ |
|---|---|---|---|
| ORE | - | 0.716 | 0.800 |
| $ATE_{trans}$ | 0.579 | - | 0.780 |
| $ATE_{rot}$ | 0.650 | 0.681 | - |

**Ranking statistics.** To verify the effectiveness of ORE, we compare rank statistics with standard metrics $ATE_{rot}$ and $ATE_{trans}$. We use two standard rank statistics: Spearman's rank correlation coefficient (Spearman Coef.) [100] and Kendall's $\tau$ coefficient (Kendall Coef.) [1]. Spearman measures Pearson correlation between rankings of two variables; values above 0.6 indicate strong correlation and 0.8 is considered very strong. Kendall measures ordinal association in a pairwise manner; values above 0.35 indicate strong association. We refer to supplementary for details.

### 4.2.1 ORE strongly correlates to other metrics

We compute ranking statistics between ORE, $ATE_{trans}$ and $ATE_{rot}$ (3) and averaged across scenes. ORE has strong correlation with both $ATE_{trans}$ and $ATE_{rot}$. Hence ORE leads to conclusions consistent with metrics computed from groundtruth. It is worth noting that the correlation is even stronger than correlation between standard metrics themselves (i.e. $ATE_{trans}$ vs. $ATE_{rot}$). Even $ATE_{trans}$ and $ATE_{rot}$ may rank methods differently, for example TartanVO has low translation error but the highest rotation error. In addition, we observe ORE correlates better with ATE, since both are considering the entire camera trajectory, not a local window as in RPE (see suppl.). Between translation and rotation, ORE correlates stronger with rotation.

*Remark*: Since ORE is a single method, we remark that rank correlations for ORE are in some sense upper bounded by the inconsistency between $ATE_{trans}$ and $ATE_{rot}$: we can agree with only one of the two if these disagree. By removing inconsistent pairs (16% of all pairs), Kendell Coef. between ORE and ATE improves to 0.895 (see suppl.).

### 4.2.2 Design choices for benchmarking ORE

**Size of bounding box.** Smaller bounding box should be more sensitive to errors (sec. 3.3). Indeed, we find higher correlation between ORE and ATE as we decrease bounding box size. Specifically, we expand bounding box sizes from 20% to 1000%, and observed decreasing Spearman Coef. and Kendall Coef. between ORE and $ATE_{rot}$/$ATE_{trans}$. This empirically verifies our intuition.
**Relationship to geometric reprojection error**. When the bounding box size reaches zero and collapses into a point, ORE effectively measures the reprojection error on this point tracklet, which was sometime referred to as (Geometric) Reprojection Error (GRE) [38, 24]. We implemented GRE metrics and found the ranking correlation to improve (albeit marginally) compared to small-sized bounding boxes (box size 0 % vs. box size 20 %). Meanwhile, we remark several practical reasons for using static object tracklets over point tracklet annotations: a. easy to track: objects may survive under occlusion, large viewpoint change and motion blur, while points may lose track; b. object annotations are more accessible than point-tracking annotations, such as Ego4D [43], EPIC-VISOR [22], etc; c. object annotations require less efforts as point annotations are extremely time consuming [25].

**Depth optimization finds optimal value.** In sec 3.2, we remark that computing ORE relies on a step to optimize for depth value that minimizes ORE. We verify that this process indeed finds the correct depth value. When groundtruth camera trajectory is used, optimal depth is almost always recovered by minimizing ORE (fig. 3a). In addition, depth value found for a worse-performing method will also

Table 4: Smaller bounding box provides stronger ranking correlation for ORE with standard metrics.

| Box Size(%) | | 1000 | 200 | 100 | 60 | 50 | 40 | 30 | 20 | 0 |
|---|---|---|---|---|---|---|---|---|---|---|
| Spearman | vs. $ATE_{trans}$ | 0.262 | 0.520 | 0.559 | 0.693 | 0.692 | 0.718 | 0.719 | 0.716 | 0.720 |
| | vs. $ATE_{rot}$ | 0.384 | 0.593 | 0.701 | 0.759 | 0.763 | 0.772 | 0.788 | 0.800 | 0.800 |
| Kendall $\tau$ | vs. $ATE_{trans}$ | 0.219 | 0.412 | 0.457 | 0.571 | 0.569 | 0.588 | 0.597 | 0.579 | 0.610 |
| | vs. $ATE_{rot}$ | 0.319 | 0.480 | 0.569 | 0.614 | 0.618 | 0.637 | 0.648 | 0.650 | 0.662 |

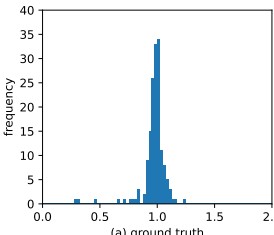 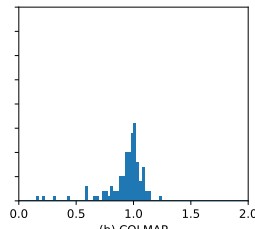 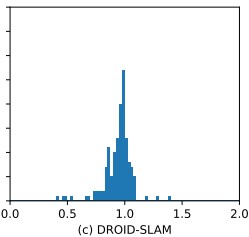

Figure 3: The ratio between optimized depth $d^*$ and ground truth depth $\hat{d}$ using (a) GT (b) COLMAP and (c) DROID trajectories. Minimizing ORE with GT trajectories almost always provide GT depth.

be less accurate. For example, minimizng ORE with COLMAP trajectories (fig. 3b) finds optimal depth in fewer sequences than better-performing DROID-SLAM (fig. 3c).

## 4.3 Experiments on EgoStatic

EgoTracks encompasses hundreds of scenarios from Ego4D and contains 6000+ unique sequences. Benchmarking on all videos from all scenes is infeasible, as many baselines may take very long time to finish (e.g. ParticleSfM and COLMAP may take up to 3 days). We believe users of EgoStatic shall choose scenarios of their interests and benchmark on a subset of sequences. Here, we select 18 most common scenarios from EgoStatic and summarize them into 11 categories, containing 30 video sequences. Each video is down-sampled to 0.25 of the original resolution (roughly 480p). We undistort the fisheye videos as most methods are designed for only pinhole camera models. We use 5 FPS following EgoTracks. We report aggregated results by taking average in each category (Table 5) and leave the per-sequence results in supplementary.

### 4.3.1 EgoStatic is challenging

As summarized in Table 5, all methods suffer from significant performance drop on EgoStatic compared to ScanNet. We remark an ORE greater than 0.2 often signals very noisy predictions (see supplementary). COLMAP is the best performing method, which indicates why it is often used in previous works requiring spatial relations in egocentric videos. However, its ORE is still relatively high, underscoring the difficulty. Like COLMAP, pure gemeotric approaches, such as ORBSLAM3, suffer from smaller performance drop. On the other hand, DROID-SLAM, the best method on ScanNet, scores 0.185. This highlights a potential difficulty for it to generalize, as it is trained on synthetic data in TartanAir [96]. In addition, we notice large variance in all methods in different sequences and categories. For example, all methods perform poorly on Social. Social often involves many moving people, increasing the difficulty to identify good static correspondences across frames.

### 4.3.2 What makes good camera trajectory predictor in EgoStatic?

In this section, we study the key factors contributing to a good camera pose estimation model for egocentric videos. Specifically, we revisit the design choices of SOTA models, COLMAP (geometry-based) and DROID-SLAM (learning-based). We demonstrate that ORE is sensitive to these factors, and therefore is a strong metric for future research.

**Design choices in COLMAP.** We study three versions of COLMAP: (a) **Default** uses the default automatic reconstruction pipeline; (b) **Tuned** replaces the exhaustive matching with vocabulary tree matching [75, 89]; (c) **Mask** filters dynamic components via hand-object masks from EgoHOS [102];

Table 5: **Egostatic is challenging: all methods suffer vs. ScanNet.** Trajectories with ORE > 0.2 are often very noisy; nearly all methods except COLMAP give very poor quality on EgoStatic.

|  | DROID | COLMAP | ORB2 | ORB3 | Particle | Monodepth2 | TartanVO |
|---|---|---|---|---|---|---|---|
| yardwork | 0.110 | 0.038 | 0.105 | 0.066 | 0.189 | 0.332 | 0.296 |
| mechanics | 0.150 | 0.073 | 0.148 | 0.163 | 0.213 | 0.271 | 0.364 |
| crafting | 0.029 | 0.006 | 0.113 | 0.082 | 0.307 | 0.486 | 0.393 |
| carpenter | 0.265 | 0.017 | 0.167 | 0.153 | 0.334 | 0.335 | 0.369 |
| lab | 0.236 | 0.018 | 0.259 | 0.088 | 0.234 | 0.421 | 0.540 |
| gardening | 0.011 | 0.009 | 0.051 | 0.072 | 0.123 | 0.200 | 0.193 |
| working | 0.255 | 0.127 | 0.192 | 0.173 | 0.194 | 0.351 | 0.456 |
| cooking | 0.315 | 0.010 | 0.177 | 0.045 | 0.179 | 0.235 | 0.299 |
| shopping | 0.165 | 0.067 | 0.171 | 0.186 | 0.227 | 0.450 | 0.396 |
| board game | 0.194 | 0.030 | 0.118 | 0.141 | 0.355 | 0.425 | 0.375 |
| social | 0.220 | 0.186 | 0.224 | 0.168 | 0.116 | 0.331 | 0.421 |
| Average | 0.185 | 0.066 | 0.161 | 0.134 | 0.225 | 0.349 | 0.383 |
| vs. ScanNet | +0.156 | +0.031 | +0.059 | +0.025 | +0.095 | +0.098 | +0.089 |

and (d) **Distortion** runs on original raw videos from Ego4D which includes severe distortion and fisheye effects. We summarized each version's performance in Table 6.

Compared to Default, Tuned achieves 0.041 ORE (-0.025). This suggests the practice introduced by [89] is indeed improving camera trajectory qualities. It is also worth noting Tuned runs about 1.5x faster than Default. Compared to Tuned, Distortion achieves 0.044 ORE (+0.003), which suggests that the internal fisheye camera model from COLMAP performs decently for in-the-wild Ego4D videos even without manual undistortion.

**Design choices in DROID-SLAM.** DROID-SLAM [87] performs recurrent iterative update operation of camera pose and pixelwise depth through a Dense Bundle Adjustment layer [87]. In the paper implementation, this update operation has been applied to both the local bundle adjustment in frontend and global bundle adjustment in backend. We used ORE metric on EgoStatic benchmark to quantify the relative gain from the addition of global bundle adjustment and see that ORE is able to capture the improvement induced by backend global bundle adjustment as in Table 7.

Table 6: Different version of COLMAP's performance on Egostatic.

|  | (a) COLMAP-Default | (b) COLMAP-Tuned | (c) COLMAP-Mask | (d) COLMAP-Distortion |
|---|---|---|---|---|
| ORE | 0.066 | 0.041 | 0.049 | 0.044 |

Table 7: Removing DROID-SLAM's global backend optimization module [87] decreases model performance on both Ego4d and ScanNet dataset.

|  | EgoStatic-ORE | ScanNet-ORE | ScanNet-ATE$_{trans}$ | ScanNet-ATE$_{rot}$ |
|---|---|---|---|---|
| Frontend and Backend | 0.185 | 0.029 | 0.147 | 0.135 |
| Frontend Only | 0.214 | 0.053 | 0.218 | 0.232 |

# 5 Discussions and limitations

We presented ORE, a metric to evaluate camera trajectories with lightweight tracking annotation. ORE removes the need to collect groundtruth camera trajectories and uses object tracklets as a proxy. By applying ORE to Ego4D, we construct EgoStatic, a new large-scale benchmark for egocentric videos. We show that estimating camera trajectories in real-world egocentric videos is very challenging: existing approaches and their simple variants do not offer satisfying solutions. We hope that ORE can give researchers the convenience to study camera poses in more diversified scenarios, and EgoStatic to serve as a new playground to improve camera localization in egocentric videos.

**Limitations.** We remark that ORE is complementary to standard metrics computed with groundtruth camera pose, and is not meant to replace them; we feel the right level of spatial detail depends on the task, and view the proposed benchmark as highly compatible with more accurate metric methods. It should serve as a way to enrich current evaluation sets with more diversity of videos, especially when groundtruth is difficult to acquire, or video capture has already occurred. As learned methods become more commonplace, increased visual diversity becomes ever more valuable. We see the current work as a step towards ways that semantics and geometry can be measured in a more coherent fashion.

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

# A   Supplementary List

We include the following contents in the supplementary materials,

1. **supplemental.pdf**: This PDF contains more experiments/ablations, visualizations as well as information about EgoStatic dataset.

2. **visualization_1.mp4**, **visualization_2.mp4** and **visualization_3.mp4**: visualization videos of EgoStatic benchmark where colored bounding boxes are groundtruth tracklet annotations, colored dots on the left are reprojected points tracked by estimated camera trajectory and green dots on the right are visualization of estimated camera trajectory (from COLMAP[33]).

3. **egostatic.zip**: This repo contains (a) code snippets for ORE evaluation on EgoStatic and (b) sample pose files from 7 baseline methods. Detailed instruction is included in **README.md**.

# B   Experiments on ScanNet (Continued)

**ScanNet** [21] is an RGB-D video dataset containing more than 1500 3D scans, each with 3D camera pose annotations and point-clouds. We pick high confidence instance mask from Mask-RCNN [47], and construct object tracklets by back-projecting masks to the point cloud. We benchmark methods on 20 test sequences using $ATE_{trans}$, $ATE_{rot}$ and ORE, and summarized the per-scene breakdown in Table 8, 9 and 10.

Table 8: **ORE** performance of 7 methods on ScanNet dataset

|  | DROID | COLMAP | ORB2 | ORB3 | Particle | TartanVO | Monodepth2 |
|---|---|---|---|---|---|---|---|
| scene0707_00 | 0.000 | 0.000 | 0.002 | 0.001 | 0.057 | 0.149 | 0.276 |
| scene0708_00 | 0.169 | 0.066 | 0.357 | 0.060 | 0.169 | 0.176 | 0.206 |
| scene0709_00 | 0.000 | 0.035 | 0.034 | 0.017 | 0.129 | 0.155 | 0.230 |
| scene0710_00 | 0.013 | 0.012 | 0.069 | 0.033 | 0.336 | 0.533 | 0.366 |
| scene0711_00 | 0.004 | 0.004 | 0.232 | - | 0.203 | 0.374 | 0.460 |
| scene0712_00 | 0.186 | 0.062 | 0.143 | 0.141 | 0.292 | 0.409 | 0.569 |
| scene0713_00 | 0.000 | 0.000 | 0.211 | 0.402 | 0.264 | 0.362 | 0.411 |
| scene0714_00 | 0.001 | 0.001 | - | 0.001 | 0.004 | 0.123 | 0.237 |
| scene0715_00 | 0.000 | 0.000 | 0.130 | 0.000 | 0.024 | 0.076 | 0.321 |
| scene0716_00 | 0.000 | 0.006 | 0.000 | 0.000 | 0.000 | 0.083 | 0.083 |
| scene0717_00 | 0.000 | 0.000 | 0.002 | - | 0.000 | 0.070 | 0.103 |
| scene0718_00 | 0.000 | 0.000 | - | 0.163 | 0.002 | 0.079 | 0.069 |
| scene0719_00 | 0.083 | 0.082 | 0.085 | 0.078 | 0.113 | 0.090 | 0.224 |
| scene0720_00 | 0.000 | 0.170 | 0.013 | 0.185 | 0.261 | 0.532 | 0.436 |
| scene0721_00 | 0.077 | 0.204 | 0.128 | 0.186 | 0.114 | 0.369 | 0.425 |
| scene0722_00 | 0.002 | 0.002 | 0.074 | 0.085 | 0.079 | 0.194 | 0.311 |
| scene0723_00 | 0.040 | 0.040 | 0.077 | 0.381 | 0.098 | 0.542 | 0.641 |
| scene0724_00 | 0.001 | 0.001 | 0.115 | 0.123 | 0.314 | 0.237 | 0.201 |
| scene0725_00 | 0.007 | 0.006 | 0.161 | - | 0.130 | 0.209 | 0.112 |
| scene0726_00 | 0.000 | 0.000 | 0.012 | 0.002 | 0.003 | 0.258 | 0.202 |
| **Average** | 0.029 | 0.035 | 0.102 | 0.109 | 0.130 | 0.251 | 0.294 |
| **STD** | 0.055 | 0.057 | 0.093 | 0.122 | 0.111 | 0.159 | 0.155 |

# C   Extended results on ranking statistics

In Section 4.2, we used two ranking statistics to capture the relationship between ORE and standard metrics: Spearman's rank correlation coefficient (Spearman Coef.) [100] and Kendall's $\tau$ coefficient (Kendall Coef.) [1].

Table 9: **ATE**$_{trans}$ performance of 7 methods on ScanNet dataset

| | DROID | COLMAP | ORB2 | ORB3 | Particle | TartanVO | Monodepth2 |
|---|---|---|---|---|---|---|---|
| scene0707_00 | 0.052 | 0.111 | 0.123 | 0.111 | 0.184 | 0.589 | 0.851 |
| scene0708_00 | 0.638 | 0.380 | 1.497 | 0.380 | 0.233 | 0.389 | 1.196 |
| scene0709_00 | 0.060 | 0.455 | 0.110 | 0.455 | 0.237 | 0.385 | 0.737 |
| scene0710_00 | 0.070 | 0.167 | 0.102 | 0.167 | 0.271 | 0.395 | 0.549 |
| scene0711_00 | 0.049 | 0.393 | 0.632 | - | 0.708 | 0.888 | 0.928 |
| scene0712_00 | 0.636 | 0.656 | 0.194 | 0.393 | 0.570 | 0.787 | 0.698 |
| scene0713_00 | 0.199 | 0.156 | 0.612 | 0.656 | 0.580 | 0.538 | 0.630 |
| scene0714_00 | 0.040 | 0.108 | - | 0.156 | 0.497 | 0.443 | 0.782 |
| scene0715_00 | 0.057 | 0.669 | 0.500 | 0.108 | 0.243 | 0.122 | 0.483 |
| scene0716_00 | 0.529 | 0.455 | 0.647 | 0.669 | 0.527 | 0.222 | 0.867 |
| scene0717_00 | 0.072 | 0.038 | 0.243 | - | 0.253 | 0.374 | 0.529 |
| scene0718_00 | 0.170 | 0.995 | - | 0.455 | 0.256 | 0.092 | 0.230 |
| scene0719_00 | 0.026 | 2.262 | 0.067 | 0.038 | 0.132 | 0.184 | 0.628 |
| scene0720_00 | 0.053 | 0.537 | 0.100 | 0.995 | 0.797 | 0.635 | 0.943 |
| scene0721_00 | 0.098 | 0.878 | 1.380 | 2.262 | 1.399 | 0.813 | 2.200 |
| scene0722_00 | 0.034 | 0.744 | 0.501 | 0.537 | 0.529 | 0.246 | 0.492 |
| scene0723_00 | 0.055 | 0.058 | 0.121 | 0.878 | 0.544 | 0.561 | 0.830 |
| scene0724_00 | 0.031 | 0.001 | 0.527 | 0.744 | 0.713 | 0.478 | 0.743 |
| scene0725_00 | 0.039 | 0.006 | 0.857 | - | 0.799 | 0.613 | 0.908 |
| scene0726_00 | 0.039 | 0.000 | 0.111 | 0.058 | 0.216 | 0.270 | 0.568 |
| **Average** | 0.147 | 0.331 | 0.462 | 0.533 | 0.484 | 0.451 | 0.790 |
| **STD** | 0.196 | 0.468 | 0.420 | 0.518 | 0.298 | 0.222 | 0.385 |

Table 10: **ATE**$_{rot}$ performance of 7 methods on ScanNet dataset

| | DROID | COLMAP | ORB2 | ORB3 | Particle | TartanVO | Monodepth2 |
|---|---|---|---|---|---|---|---|
| scene0707_00 | 0.035 | 0.066 | 0.094 | 0.097 | 0.176 | 2.403 | 1.251 |
| scene0708_00 | 0.783 | 0.079 | 1.564 | 0.164 | 0.158 | 2.271 | 1.926 |
| scene0709_00 | 0.029 | 1.134 | 0.086 | 0.178 | 0.220 | 2.265 | 1.586 |
| scene0710_00 | 0.058 | 0.057 | 0.203 | 0.231 | 0.449 | 1.704 | 2.382 |
| scene0711_00 | 0.033 | 0.204 | 1.068 | - | 0.976 | 1.962 | 2.251 |
| scene0712_00 | 0.450 | 0.829 | 0.233 | 0.308 | 1.836 | 1.918 | 2.096 |
| scene0713_00 | 0.211 | 0.077 | 1.573 | 2.527 | 0.581 | 2.299 | 2.646 |
| scene0714_00 | 0.041 | 0.055 | - | 0.147 | 0.154 | 2.118 | 1.044 |
| scene0715_00 | 0.056 | 0.100 | 1.638 | 0.105 | 0.295 | 2.105 | 0.814 |
| scene0716_00 | 0.296 | 0.784 | 1.557 | 0.238 | 0.458 | 2.156 | 1.165 |
| scene0717_00 | 0.033 | 0.071 | 0.116 | - | 0.266 | 2.018 | 1.087 |
| scene0718_00 | 0.226 | 0.225 | - | 1.899 | 1.089 | 2.049 | 0.606 |
| scene0719_00 | 0.051 | 0.058 | 0.118 | 0.063 | 0.104 | 1.998 | 1.937 |
| scene0720_00 | 0.049 | 1.438 | 0.197 | 1.995 | 0.633 | 2.155 | 2.582 |
| scene0721_00 | 0.065 | 2.824 | 1.966 | 2.747 | 0.842 | 2.038 | 1.982 |
| scene0722_00 | 0.034 | 0.037 | 0.746 | 2.407 | 0.673 | 1.945 | 1.717 |
| scene0723_00 | 0.051 | 0.056 | 0.146 | 2.551 | 0.328 | 2.069 | 2.746 |
| scene0724_00 | 0.050 | 0.135 | 2.349 | 2.359 | 1.806 | 2.020 | 1.906 |
| scene0725_00 | 0.086 | 0.591 | 2.352 | - | 1.266 | 2.343 | 2.605 |
| scene0726_00 | 0.060 | 0.134 | 0.114 | 0.084 | 0.266 | 2.081 | 2.899 |
| **Average** | 0.135 | 0.448 | 0.896 | 1.065 | 0.629 | 2.096 | 1.861 |
| **STD** | 0.184 | 0.676 | 0.830 | 1.096 | 0.513 | 0.161 | 0.669 |

**Spearman's rank correlation coefficient** Given $n$ raw scores $X_i$ and $Y_i$, ranking function $R(\cdot)$,

$$\rho_{R(x),R(Y)} = \frac{\text{cov}(R(X), R(Y))}{\sigma_{R(X)}\sigma_{R(Y)}} \tag{6}$$

where $\text{cov}(R(X), R(Y))$ is the covariance, $\sigma_{R(X}$ and $\sigma_{R(Y)}$ are standard deviations.

We compute Spearman's ranking statistics on each scene in Table 11, then average across all scenes.

Table 11: **Spearman Coef. and Kendall Coef. for ScanNet**. Here we show Spearman and Kendall Coef. between (a) ORE and $\text{ATE}_{trans}$, (b) ORE and $\text{ATE}_{rot}$, as well as (c) $\text{ATE}_{trans}$ and $\text{ATE}_{rot}$ for each scene in ScanNet test set. The table is summarized as Table 3 of main paper.

| ORE | Spearman Coef. | | | Kendall Coef. | | |
| | vs. $\text{ATE}_{trans}$ | vs. $\text{ATE}_{rot}$ | $\text{ATE}_{trans}$ vs $\text{ATE}_{rot}$ | vs. $\text{ATE}_{trans}$ | vs. $\text{ATE}_{rot}$ | $\text{ATE}_{trans}$ vs $\text{ATE}_{rot}$ |
|---|---|---|---|---|---|---|
| scene0707_00 | 0.964 | 0.893 | 0.929 | 0.905 | 0.714 | 0.810 |
| scene0708_00 | 0.857 | 0.750 | 0.750 | 0.714 | 0.524 | 0.619 |
| scene0709_00 | 0.607 | 0.893 | 0.714 | 0.524 | 0.714 | 0.619 |
| scene0710_00 | 0.893 | 0.929 | 0.964 | 0.714 | 0.810 | 0.905 |
| scene0711_00 | 0.964 | 1.000 | 0.964 | 0.905 | 1.000 | 0.905 |
| scene0712_00 | 0.643 | 0.750 | 0.821 | 0.429 | 0.619 | 0.619 |
| scene0713_00 | 0.786 | 0.929 | 0.857 | 0.524 | 0.810 | 0.714 |
| scene0714_00 | 0.857 | 0.857 | 0.893 | 0.714 | 0.714 | 0.810 |
| scene0715_00 | 0.889 | 0.815 | 0.786 | 0.720 | 0.617 | 0.714 |
| scene0716_00 | 0.039 | 0.670 | -0.214 | 0.056 | 0.620 | -0.238 |
| scene0717_00 | 0.889 | 0.852 | 0.929 | 0.823 | 0.720 | 0.810 |
| scene0718_00 | 0.721 | 0.919 | 0.643 | 0.586 | 0.781 | 0.619 |
| scene0719_00 | 0.821 | 0.679 | 0.929 | 0.619 | 0.429 | 0.810 |
| scene0720_00 | 0.679 | 0.857 | 0.750 | 0.524 | 0.714 | 0.619 |
| scene0721_00 | 0.464 | 0.679 | 0.607 | 0.333 | 0.524 | 0.429 |
| scene0722_00 | 0.464 | 0.821 | 0.643 | 0.429 | 0.619 | 0.429 |
| scene0723_00 | 0.857 | 0.929 | 0.964 | 0.714 | 0.810 | 0.905 |
| scene0724_00 | 0.536 | 0.321 | 0.714 | 0.333 | 0.143 | 0.619 |
| scene0725_00 | 0.643 | 0.714 | 0.964 | 0.429 | 0.524 | 0.905 |
| scene0726_00 | 0.739 | 0.739 | 1.000 | 0.586 | 0.586 | 1.000 |
| **Average (STD)** | 0.716 ($\pm$ 0.216) | 0.800 ($\pm$ 0.145) | 0.780 ($\pm$ 0.216) | 0.579 ($\pm$ 0.205) | 0.650 ($\pm$ 0.173) | 0.681 ($\pm$ 0.261) |

**Kendall's $\tau$ rank correlation coefficient** measures for ordinal association between two sets of data. Given $\{R(X_i)\}_{i=1}^n$ and $\{R(Y_i)\}_{i=1}^n$,

a pair of random variables $(X, Y)$, where $X = \{x_1, x_2, ...x_n\}$ and $Y = \{y_1, y_2, ...y_n\}$. If either both $x_i > x_j$ and $y_i > y_j$ or both $x_i < x_j$ and $y_i < y_j$ holds, the pair $(x_i, y_i)$ and $(x_j, y_j)$ are said to be concordant; otherwise they are said to be discordant. Assume for random variable $(X, Y)$, there exists $A$ concordant pairs and $B$ discordant pairs, the Kendall $\tau$ coefficient is defined as $\tau = \frac{A - B}{\binom{n}{2}}$.

We consider $(X, Y)$ as the performance of two different methods on the same scene, resulting in 420 such pairs across 20 scenes in ScanNet.

## C.1 Relationship with RPE

We summarize the ranking statistics between ORE and RPE in Supp. Tab 12. We observe despite ORE has strong correlation with RPE, it is weaker than the correlation with ATE. This is expected, since both ORE and ATE measure the quality of the entire trajectory, whereas RPE measures a small local window. In addition, similar to ATE, we observe ORE to have stronger correlation with rotation compared to translation.

## C.2 Inconsistent pairs removal in Kendall's $\tau$

As discussed in the section 4.2 in the main paper, standard metrics $ATE_{trans}$ and $ATE_{rot}$ may often disagree with each other. This indicates for two methods, one may perform better on translation while the other performs better on rotation. This limits ORE's ranking statistic values to have an upper bound: since ORE factors in both translation and rotation, it can agree with either $ATE_{trans}$ or $ATE_{rot}$ when $ATE_{trans}$ and $ATE_{rot}$ mismatch. We capture this problem by removing such pairs in Kendall's $\tau$. In total, we remove 16% of such pairs.

As summarized in Supp. Tab. 13, ORE and ATE has extremely high Kendall ranking coefficients. This implies only 5% of the pairs may disagree between ORE and ATE. ORE also acheives strong correlation with RPE, but is weaker than the global ATE measurement.

Table 12: ORE's ranking correlation with RPE. Spearman Coef. (upper triangle) and Kendall Coef. (lower triangle) between ORE and standard metrics.

| Spearman / Kendall | ORE | $RPE_{trans}$ | $RPE_{rot}$ |
|---|---|---|---|
| ORE | - | 0.425 | 0.750 |
| $RPE_{trans}$ | 0.334 | - | 0.473 |
| $RPE_{rot}$ | 0.615 | 0.386 | - |

Table 13: ORE acheives very strong correlation with ATE and RPE with inconsistent pairs removal.

| | ATE | RPE |
|---|---|---|
| Kendall | 0.895 | 0.571 |

## D  Visualization

In this section, we provided more visualization of the ORE metrics and its associated EgoStatic benchmark.
**Visualization for ORE metrics** For each of the 7 baseline methods, we plot the estimated trajectory along with the ground truth trajectory, as well as its associated ORE, $ATE_{trans}$ and $ATE_{rot}$ metric. From Figure 4, we can see both qualitatively and quantitatively that ORE correlated highly with the quality of estimated camera trajectory.
**Visualization for EgoStatic Benchmark** In Figure 5, we visualized 24 sampled screenshot from 12 arbitrary scenes in EgoStatic benchmark (each with 2 images). The scenarios span across various categories including cooking, crafting, yardwork, gardening, lab, board game etc, which demonstrate the diverse scenes and complex actions EgoStatic is able to cover.

## E  Annotation Limitations

The annotation for whether an object is static was done by human labelers. Although quality check has been placed for all jobs, it is prune to human bias and error. Also, the videos are sourced from Ego4D VQ benchmark, so it may not represent the entire data disribution of Ego4D.

## F  Potential negative societal impacts

Since our videos are sourced from Ego4D and EgoTracks, we inherit many of the potential negative impacts. Details are discussed in the original paper [43] Appendix K. We summarize several here:

- There may be risks surrounding privacy, such as personnel being recorded during the video. Consent was obtained in the original dataset, and a user agreement is enforced for Ego4D. Our dataset follow the same protocol as Ego4D.

- The existing efforts may inspire future data collection with less attention to privacy and ethics. Best practices were detailed in the original paper [43]. We did the same with our paper to include the instructions to help mitigate this risk.

- There may be data imbalances, such as geographical distribution. This risk can be mitigated with future work that grows the collaboration in underrepresented areas.

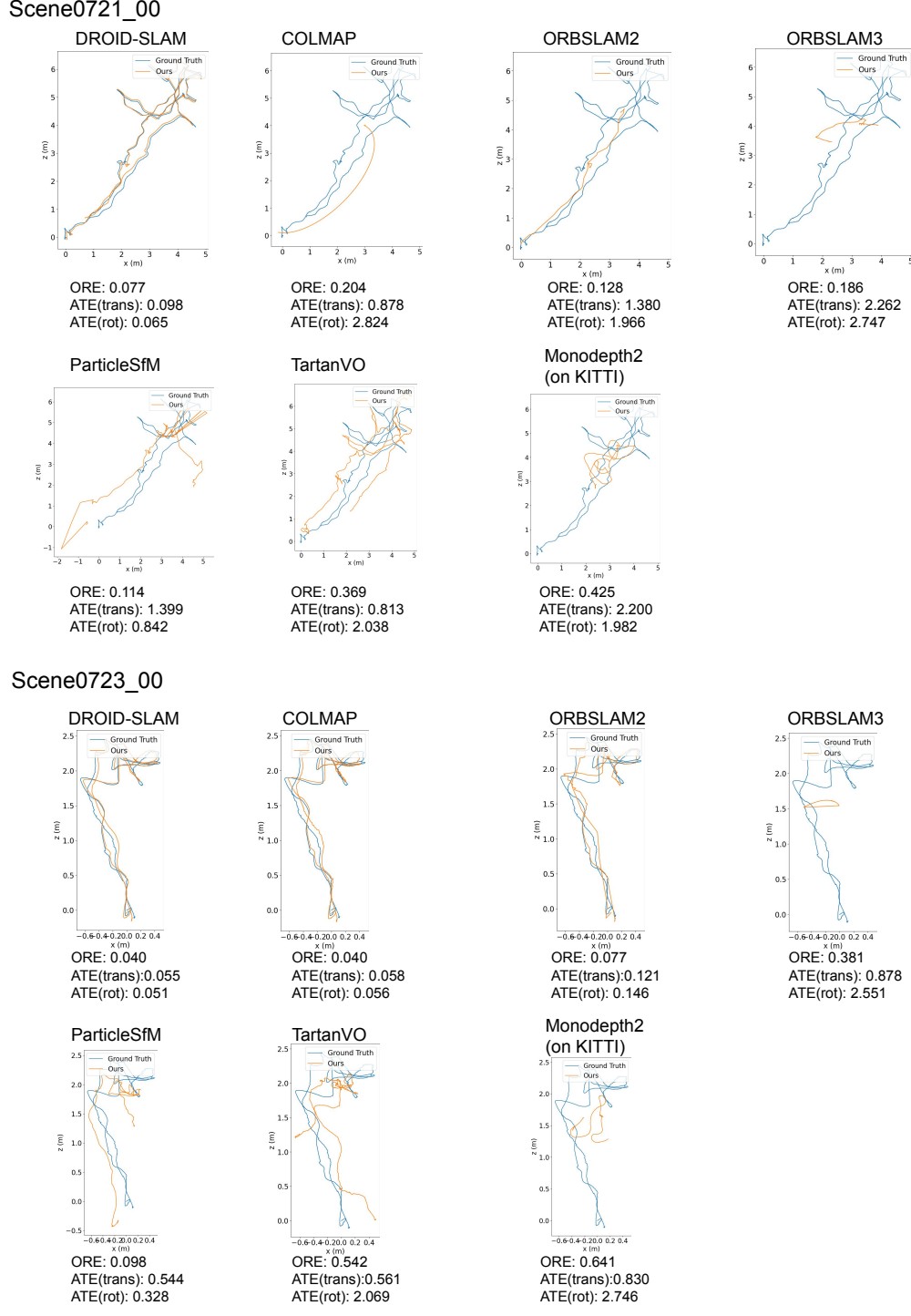

Figure 4: **Visualization of the camera trajectory** The ORE, ATE$_{trans}$ and ATE$_{rot}$ metric of 7 methods discussed in Sec 4.1 are listed below the trajectory to show that ORE highly correlates with ATE$_{trans}$ and ATE$_{rot}$ in its capability to describe the trajectory quality.

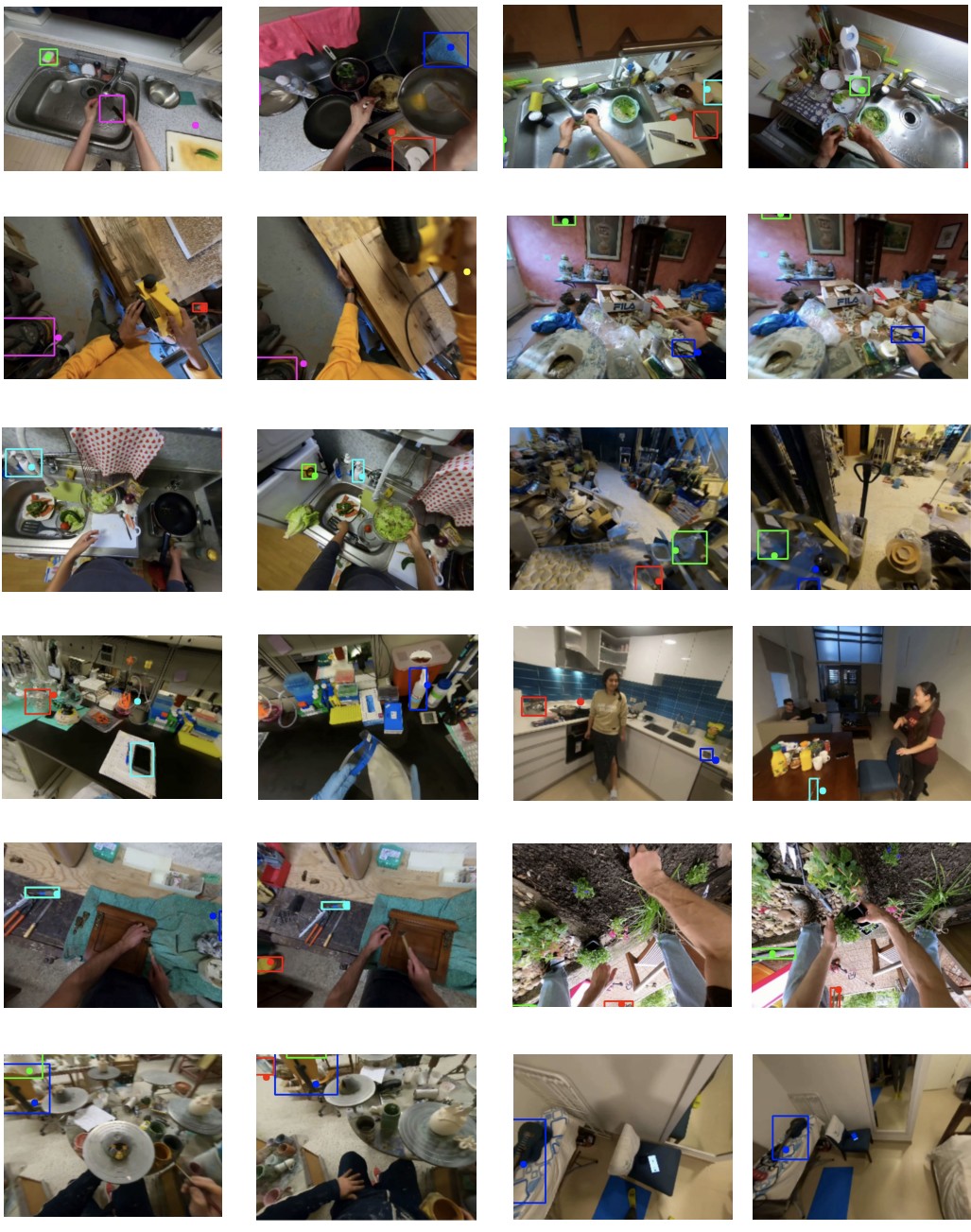

Figure 5: **Visualization of EgoStatic benchmark** We include 24 screenshots from 12 videos in the proposed EgoStatic benchmark. Colored bounding boxes are ground truth tracklet annotations; dots are reprojected points tracked by estimated camera trajectory. Ideally, the points should fall in the bounding boxes.

# G   Data sheet

We follow [35] for writing the data sheet of EgoStatic.

1. Motivation

   (a) *For what purpose was the dataset created? (Was there a specific task in mind? Was there a specific gap that needed to be filled? Please provide a description.)*
   3D spatial understanding is highly valuable in the context of semantic modeling of environments, agents, and their relationships. Semantic modeling approaches employed on monocular video often ingest outputs from off-the-shelf SLAM/SfM pipelines, which are anecdotally observed to perform poorly or fail completely on some fraction of the videos of interest. These target videos may vary widely in complexity of scenes, activities, camera trajectory, etc. Unfortunately, such semantically-rich video data often comes with no ground-truth 3D information, and in practice it is prohibitively costly or impossible to obtain ground truth reconstructions or camera pose post-hoc.
   This paper proposes a novel evaluation protocol, Object Reprojection Error (ORE) to benchmark camera trajectories; ORE computes reprojection error for static objects within the video and requires only lightweight object tracklet annotations. These annotations are easy to gather on new or existing video, enabling ORE to be calculated on essentially arbitrary datasets. We show that ORE maintains high rank correlation with standard metrics based on ground truth. Leveraging ORE, we source videos and annotations from Ego4D-EgoTracks, resulting in EgoStatic, a large-scale diverse dataset for evaluating camera trajectories in-the-wild.

   (b) *Who created this dataset (e.g., which team, research group) and on behalf of which entity (e.g., company, institution, organization)?* We prefer to stay anonymous in this submission. We refer to Ego4D [43] and EgoTracks [86] for information on dataset collection and annotations of bounding boxes. We will release this information upon paper acceptance.

   (c) *Who funded the creation of the dataset? (If there is an associated grant, please provide the name of the grantor and the grant name and number.)* We prefer to stay anonymous in this submission. We will release this information upon paper acceptance.

   (d) *Any other comments?*
   None.

2. Composition

   (a) *What do the instances that comprise the dataset represent (e.g., documents, photos, people, countries)? (Are there multiple types of instances (e.g., movies, users, and ratings; people and interactions between them; nodes and edges)? Please provide a description.)*
   Each instance is a video; our annotations are labeling object tracklets from EgoTracks with a static vs. non-static label.

   (b) *How many instances are there in total (of each type, if appropriate)?*
   There are 5708 instances in total.

   (c) *Does the dataset contain all possible instances or is it a sample (not necessarily random) of instances from a larger set? (If the dataset is a sample, then what is the larger set? Is the sample representative of the larger set (e.g., geographic coverage)? If so, please describe how this representativeness was validated/verified. If it is not representative of the larger set, please describe why not (e.g., to cover a more diverse range of instances, because instances were withheld or unavailable).)*
   The dataset does not contain all videos in Ego4D. Instead, we only have videos that are present in the Ego4D VQ benchmark [43]. Since these videos are the same, we share the same human and geographic bias and limitations in video selection.

   (d) *What data does each instance consist of? ("Raw" data (e.g., unprocessed text or images) or features? In either case, please provide a description.)*
   Each object tracklet in a video is labeled as "static" or not.

   (e) *Is there a label or target associated with each instance? If so, please provide a description.*
   Yes, see answer above.

   (f) *Is any information missing from individual instances? (If so, please provide a description, explaining why this information is missing (e.g., because it was unavailable). This does not include intentionally removed information, but might include, e.g., redacted text.)*

No, we do not remove any information for individual instances already present in Ego4D [43].

(g) *Are relationships between individual instances made explicit (e.g., users' movie ratings, social network links)? ( If so, please describe how these relationships are made explicit.)*

No. Since videos in Ego4D can be hours long, we follow the same procedure to split a video into multiple shorter video clips [43]. Some videos clips may may have been captured by the same individual, but we do not expose this information in the dataset.

(h) *Are there recommended data splits (e.g., training, development/validation, testing)? (If so, please provide a description of these splits, explaining the rationale behind them.)*

No. The benchmark is mainly designed for evaluation. However, given the large-scale of this benchmark, one may leverage this to train models. We leave this as promising future works.

(i) *Are there any errors, sources of noise, or redundancies in the dataset? (If so, please provide a description.)*

Since the labels are annotated by human raters, they are prone to human bias and errors. We applied quality assurance procedures to minimize such errors where we can.

(j) *Is the dataset self-contained, or does it link to or otherwise rely on external resources (e.g., websites, tweets, other datasets)? (If it links to or relies on external resources, a) are there guarantees that they will exist, and remain constant, over time; b) are there official archival versions of the complete dataset (i.e., including the external resources as they existed at the time the dataset was created); c) are there any restrictions (e.g., licenses, fees) associated with any of the external resources that might apply to a future user? Please provide descriptions of all external resources and any restrictions associated with them, as well as links or other access points, as appropriate.)*

The underlying videos are sourced from [43]. The [43] dataset is maintained by Ego4D consortium which can be regarded as guaranteed to exist and remain constant. The license can be found `https://ego4d-data.org/pdfs/Ego4D-Licenses-Draft.pdf`.

(k) *Does the dataset contain data that might be considered confidential (e.g., data that is protected by legal privilege or by doctor-patient confidentiality, data that includes the content of individuals' non-public communications)? (If so, please provide a description.)*

No. Ego4D [43] was collected with careful consideration of ethics and consent. We largely inherit these characteristics. The additional attribute ("static") annotations of our dataset do not reveal any confidential information.

(l) *Does the dataset contain data that, if viewed directly, might be offensive, insulting, threatening, or might otherwise cause anxiety? (If so, please describe why.)*

We share the same underlying videos as [43] and objects as [86], so we inherit its privacy and ethics standards, as well as any potential risks. The additional attribute annotations of our dataset do not add any additional objectionable content.

(m) *Does the dataset relate to people? (If not, you may skip the remaining questions in this section.)*

Yes. The annotations we provide do not contain any information relating to people; however, they are labeled from Ego4D [43] videos, which do contain people, and indirectly are related to people due to its egocentric nature.

(n) *Does the dataset identify any subpopulations (e.g., by age, gender)? (If so, please describe how these subpopulations are identified and provide a description of their respective distributions within the dataset.)*

EgoStatic does not identify subpopulations, but the underlying dataset Ego4D [43] does. See Appendix C of [43] for more details.

(o) *Is it possible to identify individuals (i.e., one or more natural persons), either directly or indirectly (i.e., in combination with other data) from the dataset? (If so, please describe how.)*

EgoStatic does not contain tracks of people, focusing instead on objects. Ego4D [43] does contain some visually identifiable individuals, who provided their consent to appear in the dataset. Other individuals have their faces blurred.

(p) *Does the dataset contain data that might be considered sensitive in any way (e.g., data that reveals racial or ethnic origins, sexual orientations, religious beliefs, political opinions or union memberships, or locations; financial or health data; biometric or genetic data; forms of government identification, such as social security numbers; criminal history)? (If so, please provide a description.)*

EgoStatic does not add any labels for people, focusing instead on static objects. Ego4D [43] does contain demographics information. See (n) above.

(q) *Any other comments?*

N/A.

3. Collection Process

(a) *How was the data associated with each instance acquired? (Was the data directly observable (e.g., raw text, movie ratings), reported by subjects (e.g., survey responses), or indirectly inferred/derived from other data (e.g., part-of-speech tags, model-based guesses for age or language)? If data was reported by subjects or indirectly inferred/derived from other data, was the data validated/verified? If so, please describe how.)*

The attribute annotation for each object track was acquired by human annotators. They were instructed to label a track as "static" or not.

(b) *What mechanisms or procedures were used to collect the data (e.g., hardware apparatus or sensor, manual human curation, software program, software API)? (How were these mechanisms or procedures validated?)*

We use the proprietary annotation software to collect "static" attribute annotations. The software shows a video frame by frame, and the annotator is able to select from a dropdown menu whether the object is "static". The software will then record the response for each object track.

(c) *If the dataset is a sample from a larger set, what was the sampling strategy (e.g., deterministic, probabilistic with specific sampling probabilities)?*

We used the entirety of Ego4D's Visual Queries benchmark [43] and object tracks from EgoTracks [86] as the basis of EgoStatic.

(d) *Who was involved in the data collection process (e.g., students, crowdworkers, contractors) and how were they compensated (e.g., how much were crowdworkers paid)?*

The participants were contractors employed by a third-party vendor and are compensated based on the agreement with their employer.

(e) *Over what timeframe was the data collected? (Does this timeframe match the creation timeframe of the data associated with the instances (e.g., recent crawl of old news articles)? If not, please describe the timeframe in which the data associated with the instances was created.)*

The dataset was created in winter and spring of 2023, which is not the time the videos were collected.

(f) *Were any ethical review processes conducted (e.g., by an institutional review board)? (If so, please provide a description of these review processes, including the outcomes, as well as a link or other access point to any supporting documentation.)*

Yes. For proprietary reasons, we are not able to provide supporting documentation. Our internal review was conducted thoroughly vetted potential privacy and ethical related concerns.

(g) *Does the dataset relate to people? (If not, you may skip the remaining questions in this section.)*

Yes. The annotations we provide in EgoStatic do not contain any information relating to people; however, they are labeled from Ego4D [43] videos, which do contain people, and indirectly are related to people due to its egocentric nature.

(h) *Did you collect the data from the individuals in question directly, or obtain it via third parties or other sources (e.g., websites)?*

Ego4D [43] collected the data from the individual directly. We do not collect any additional videos beyond those in Ego4D.

(i) *Were the individuals in question notified about the data collection? (If so, please describe (or show with screenshots or other information) how notice was provided, and*

*provide a link or other access point to, or otherwise reproduce, the exact language of the notification itself.)*

Ego4D [43] was collected by willing and consenting individuals. See Section 3.4 in [43].

(j) *Did the individuals in question consent to the collection and use of their data? (If so, please describe (or show with screenshots or other information) how consent was requested and provided, and provide a link or other access point to, or otherwise reproduce, the exact language to which the individuals consented.)*

Ego4D [43] was collected by willing and consenting individuals. See Section 3.4 in [43].

(k) *If consent was obtained, were the consenting individuals provided with a mechanism to revoke their consent in the future or for certain uses? (If so, please provide a description, as well as a link or other access point to the mechanism (if appropriate).)*

Yes, see Section 3.4 in [43].

(l) *Has an analysis of the potential impact of the dataset and its use on data subjects (e.g., a data protection impact analysis) been conducted? (If so, please provide a description of this analysis, including the outcomes, as well as a link or other access point to any supporting documentation.)*

No.

(m) *Any other comments?*

None.

4. Preprocessing/cleaning/labeling

(a) *Was any preprocessing/cleaning/labeling of the data done (e.g., discretization or bucketing, tokenization, part-of-speech tagging, SIFT feature extraction, removal of instances, processing of missing values)? (If so, please provide a description. If not, you may skip the remainder of the questions in this section.)*

No.

(b) *Was the "raw" data saved in addition to the preprocessed/cleaned/labeled data (e.g., to support unanticipated future uses)? (If so, please provide a link or other access point to the "raw" data.)*

No.

(c) *Is the software used to preprocess/clean/label the instances available? (If so, please provide a link or other access point.)*

No.

(d) *Any other comments?*

None.

5. Uses

(a) *Has the dataset been used for any tasks already? (If so, please provide a description.)*

The dataset has been used for understanding the performance of SLAM methods.

(b) *Is there a repository that links to any or all papers or systems that use the dataset? (If so, please provide a link or other access point.)*

N/A.

(c) *What (other) tasks could the dataset be used for?*

The dataset could possibly be used for training SLAM system that focus on the semantic relationships between objects.

(d) *Is there anything about the composition of the dataset or the way it was collected and preprocessed/cleaned/labeled that might impact future uses? (For example, is there anything that a future user might need to know to avoid uses that could result in unfair treatment of individuals or groups (e.g., stereotyping, quality of service issues) or other undesirable harms (e.g., financial harms, legal risks) If so, please provide a description. Is there anything a future user could do to mitigate these undesirable harms?)*

The selection of objects is inherited from the Ego4D visual-query benchmark and Ego-Tracks, which is biased towards objects with appearances of at least modest duration. Consequently, our benchmark may contain less instances that only appear very briefly in the video.

(e) *Are there tasks for which the dataset should not be used? (If so, please provide a description.)*
We currently do not foresee anything such tasks.

(f) *Any other comments?*
None.

6. Distribution

(a) *Will the dataset be distributed to third parties outside of the entity (e.g., company, institution, organization) on behalf of which the dataset was created? (If so, please provide a description.)*
Yes, we will share the same license and distribution as [43].

(b) *How will the dataset will be distributed (e.g., tarball on website, API, GitHub)? (Does the dataset have a digital object identifier (DOI)?)*
The dataset will be distributed via GitHub.

(c) *When will the dataset be distributed?*
Upon acceptance of this work.

(d) *Will the dataset be distributed under a copyright or other intellectual property (IP) license, and/or under applicable terms of use (ToU)? (If so, please describe this license and/or ToU, and provide a link or other access point to, or otherwise reproduce, any relevant licensing terms or ToU, as well as any fees associated with these restrictions.)*
The dataset will share the same license as Ego4D. Please see `https://ego4d-data.org/pdfs/Ego4D-Licenses-Draft.pdf`.

(e) *Have any third parties imposed IP-based or other restrictions on the data associated with the instances? (If so, please describe these restrictions, and provide a link or other access point to, or otherwise reproduce, any relevant licensing terms, as well as any fees associated with these restrictions.)*
Not to our knowledge.

(f) *Do any export controls or other regulatory restrictions apply to the dataset or to individual instances? (If so, please describe these restrictions, and provide a link or other access point to, or otherwise reproduce, any supporting documentation.)*
Not to our knowledge.

(g) *Any other comments?*
None.

7. Maintenance

(a) *Who is supporting/hosting/maintaining the dataset?*
All authors will maintain the dataset.

(b) *How can the owner/curator/manager of the dataset be contacted (e.g., email address)?*
Authors can be contacted via emails.

(c) *Is there an erratum? (If so, please provide a link or other access point.)*
Not currently. Future versions of the dataset may be released if we find errors, which will be provided within the same GitHub.

(d) *Will the dataset be updated (e.g., to correct labeling errors, add new instances, delete instances')? (If so, please describe how often, by whom, and how updates will be communicated to users (e.g., mailing list, GitHub)?)*
See previous question.

(e) *If the dataset relates to people, are there applicable limits on the retention of the data associated with the instances (e.g., were individuals in question told that their data would be retained for a fixed period of time and then deleted)? (If so, please describe these limits and explain how they will be enforced.)*
N/A. The dataset is not related to people.

(f) *Will older versions of the dataset continue to be supported/hosted/maintained? (If so, please describe how. If not, please describe how its obsolescence will be communicated to users.)*
Yes, all data will be versioned.

(g) *If others want to extend/augment/build on/contribute to the dataset, is there a mecha-*
*nism for them to do so? (If so, please provide a description. Will these contributions be*
*validated/verified? If so, please describe how. If not, why not? Is there a process for*
*communicating/distributing these contributions to other users? If so, please provide a*
*description.)*
Errors/features can be submitted as issues/pull requests on GitHub.

(h) *Any other comments?*
None.

