# OpenReview forum: "Object Reprojection Error (ORE): Camera pose benchmarks from lightweight tracking annotations"
_NeurIPS.cc/2023/Track/Datasets_and_Benchmarks — NeurIPS 2023 Datasets and Benchmarks Poster_

### Official Review · Reviewer_C25m · 2023-06-24
**A New Evaluation Protocol for Camera Trajectory Estimation**

**Rating:** 7
**Confidence:** 3
**Clarity:** Yes

**Strengths:**

(1) The insights from the research work are properly stated. This work has clearly shown the impact of the proposed protocol for camera trajectory Evaluation. (2) The authors did analyze ablation experiments (Table 4).

**Additional Feedback:**

N/A

**Correctness:**

No. The authors mention that the proposed method is a new protocol, but do not provide access to the demo code or any other documentation.

**Documentation:**

No. A demo code with a tutorial video should be posted on GitHub for readers.

**Ethics:**

No ethics issues.

**Limitations:**

(1) Page 2: Please use a scalebar beside Figure 2 to understand better spatial variations. -- (2) Page 4, Equation 3: How to select a parameter for Delta (fixed interval)? -- (3) Page 9, Discussion: How sensitive is the algorithm towards outliers (numerical results)? Prove an analysis to justify any findings.

**Opportunities For Improvement:**

Experiments: More evaluation needs to be done to understand the mechanics behind the improvement which could be extended to future work. More and new datasets, could be used to verify the claimed advantages of your methods (e.g., drone-based time-series images)?

**Relation To Prior Work:**

Yes

**Summary And Contributions:**

This manuscript presents an interesting protocol for the performance improvement of camera trajectory from lightweight tracking annotations based on time-series images without a camera trajectory truth map. The idea of the manuscript is good, and I feel that the manuscript has potential.

---

> ### Author Response · Authors · 2023-08-15
> **Response to Reviewer C25m**
>
> > Q1: Experiments: More evaluation needs to be done to understand the mechanics behind the improvement which could be extended to future work. More and new datasets, could be used to verify the claimed advantages of your methods (e.g., drone-based time-series images)?
>
> A1: Thanks for suggesting this. We note that the improvements of MonoDepth2 came mainly from finetuning the method. We further finetune the model on Ego4D videos and found it as a useful test-time finetuning strategy but may not generalize to other videos. This reconfirms our findings that simple techniques, such as finetuning, may not be sufficient to solve the problem. For other improvements we adopted to COLMAP, we clarify that our goal is not to propose these, since they were used in previous works NeuralDiff [94] ParticleSfM [95]. However, quantitative evaluation of these improvements have not been obtained due to the lack of benchmarking on egocentric videos, and EgoStatic is presented to bridge this gap to quantify the gains from these tricks.
>
> [95] Zhao, Wang, et al. "Particlesfm: Exploiting dense point trajectories for localizing moving cameras in the wild." European Conference on Computer Vision. Cham: Springer Nature Switzerland, 2022.
>
> > Q2: (1) Page 2: Please use a scalebar beside Figure 2 to understand better spatial variations. -- (2) Page 4, Equation 3: How to select a parameter for Delta (fixed interval)? -- (3) Page 9, Discussion: How sensitive is the algorithm towards outliers (numerical results)? Prove an analysis to justify any findings.
>
> A2: Thanks for the comments. We will address them accordingly in the updated version.
> For (2), delta is an internal parameter for RPE (relative pose error), a standard metric for camera pose estimation. In ScanNet experiment from Supplemental material, we follow KITTI evaluation protocol used in [96] and set delta = 1 to find framewise relative pose error.
>
> [96] Zhan, Huangying, et al. "Visual odometry revisited: What should be learnt?." 2020 IEEE international conference on robotics and automation (ICRA). IEEE, 2020.
>
> For (3), we would like to refer the reviewers to our study in Table 4 on how sensitive ORE is to errors. Specifically, if the bounding box fails to localize the object tightly, error in annotation will hurt the ORE’s capability in benchmarking the methods and thus decrease the ranking correlation with classic metrics like ATE or RPE.
> We will be happy to provide more experiments if the reviewer could clarify on what other type of outliers we can benchmark with.
>
> > Q3: Regarding documentation:
>
> A3: We have provided code for evaluating ORE in the supplementary materials. The code will be released publicly upon paper acceptance.

---

> > ### Comment · Reviewer_C25m · 2023-08-17
> >
> > All the comments were well met.

---

### Official Review · Reviewer_u9zJ · 2023-07-24

**Rating:** 6
**Confidence:** 5
**Correctness:** Those two points raised in improvemen…
**Clarity:** Clear to me.

**Strengths:**

+ The problem of interest is important. And the proposed method (using object reprojection error) makes some sense.
+ Evaluations using typical SLAM/SfM methods (including Colmap and Droid-SLAM, which is interesting) show that ORE correlates well with APE.
+ Other experiments analyze the impact of box size and domain adaptation give more value.

**Additional Feedback:**

None.

**Documentation:**

Not released yet.

**Limitations:**

No non-technical concerns.

**Opportunities For Improvement:**

+ I have concerns w.r.t. the object spatial memory motivation part. The authors even say that this study falls beween SLAM and scene graph generation. In my opinion, this is far too overclaiming. This study shows no results about spatial memroy (its construction or usage). I suggest rephrasing these fancy words with more grounded arguments about SLAM evaluation.
+ I have concerns w.r.t. the ultimate utility. The authors claim the metric can be used to create better algorithms. They use two experiments 'Design Choices in Colmap' and 'Domain adaptation of Monodepth2'. But as far as I can see, these are still showing the correlation between existsing ablations and ORE instead of truly digging out some improvements using ORE.
+ The authors now focus on SLAM/SfM and I suggest some references on relocalization (if not benchmarked), like [A] and [B].

[A] Sc-wls: Towards interpretable feed-forward camera re-localization, ECCV 2022

[B] Dfnet: Enhance absolute pose regression with direct feature matching, ECCV 2022

**Relation To Prior Work:**

Better have some references on relocalization.

**Summary And Contributions:**

This submission studies the problem of benchmarking SLAM/SfM algorithms on egocentric videos, where gt camera poses are difficult to obtain. This is an important problem as conventional SLAM benchmarks are small in scale, synthetic or using pseudo gt. The idea is to unproject object tracklets into 3D using captured depth or a joint alignment procedure, then check the re-projection consistency. The authors use an exiting dataset with 2D tracklets and describe rules to fidn out good sequences for ORE benchmarking. Seven representative methods are studied, showing that ORE well reasonates with APE. An argument is made that ORE can be used to hyperparameter selection but seems not well supported.

---

> ### Author Response · Authors · 2023-08-15
> **Response to Reviewer u9zJ**
>
> > Q1: I have concerns w.r.t. the object spatial memory motivation part. The authors even say that this study falls beween SLAM and scene graph generation. In my opinion, this is far too overclaiming. This study shows no results about spatial memroy (its construction or usage). I suggest rephrasing these fancy words with more grounded arguments about SLAM evaluation.
>
> A1: We apologize for any confusion; we invoked semantics and scene graph ideas as motivation for an object-centric benchmark for 3D localization, not to position our work as being comparable to scene graphs in terms of semantic complexity.  We see significant value in bringing 3D understanding to real-world semantic tasks, and we wanted to emphasize this set of applications.  We view ORE as a largely geometric contribution which emphasizes landmarks of semantic interest, and we believe this agrees with the reviewer’s feeling.  We will adjust how we position the dataset in “Contributions” to be more concrete, and review language elsewhere in the introduction.
>
> Our intended revision is “As a measure of environmental understanding, the resulting benchmark uses a type of reprojection error which is a generalization of geometric reprojection error; in this way it is similar to traditional SfM / SLAM benchmarks.  However, using a sparse set of semantic landmarks, specifically static objects whose identities are unknown to the method under evaluation, keeps the benchmark focused on high-level percepts in the vicinity of the camera, vs. global maps and reconstruction accuracy.”
>
> We are happy to reflect any further concerns that reviewer might have.
>
> > Q2: I have concerns w.r.t. the ultimate utility. The authors claim the metric can be used to create better algorithms. They use two experiments 'Design Choices in Colmap' and 'Domain adaptation of Monodepth2'. But as far as I can see, these are still showing the correlation between existsing ablations and ORE instead of truly digging out some improvements using ORE.
>
> A2: We would like to clarify that the main contribution of the paper is the evaluation protocol ORE and benchmark EgoStatic we developed to benchmark camera trajectories using lightweight tracking annotation. The goal of 4.3.2 is to validate the capabilities of ORE when used as a diagnostic tool, to identify the best improvements. For the COLMAP experiments, we took a number of improvements typically used by practitioners (based on empirical findings, but not quantified due to lack of ground truth), and showed that indeed ORE is capable of identifying these settings as the optimal choices. For the MonoDepth2 experiments, we show that finetuning on NYUv2 improves on ScanNet on ORE; and this improvement is cross-validated with improvements on ATE_trans and ATE_rot.
>
> We believe that proposing new methodology directions is out of the scope for a dataset/ benchmark paper. Nevertheless, studying with ORE on EgoStatic reveals new yet somewhat surprising results. For example, as pointed out by Reviewer 2WCT, COLMAP with the appropriate hyperparameter settings achieves superior performance than data-driven methods. In addition, we finetuned MonoDepth2 on Ego4D videos, and show that test-time finetuning can improve the performance, while it hurts generalization to other videos in EgoStatic (see Q3 answer for Reviewer 2WCT). We believe that EgoStatic can be used in future works as the testbed for new methods since it can robustly capture the improvements quantitatively.
>
> > Q3: The authors now focus on SLAM/SfM and I suggest some references on relocalization (if not benchmarked), like [A] and [B].
>
> Thanks reviewer for bringing this line of work to our attention. We will include the reference and discussion in the related work. ORE and EgoStatic are designed to benchmark SLAM/SfM methods; on the other hand, these re-localization methods are not directly benchmarkable on EgoStatic using ORE since [A] and [B] requires test-time training with ground truth poses or 3D reconstructions.
>
> > Q4: Regarding documentation:
>
> A3: We have provided code for evaluating ORE in the supplementary materials. The code will be released publicly upon paper acceptance.

---

> > ### Comment · Reviewer_u9zJ · 2023-08-20
> >
> > Since the dataset track allows an update to the PDF, could the authors please incorporate these intended changes into the file?

---

> > > ### Author Response · Authors · 2023-08-22
> > >
> > > Thanks for the comments. We have revised the paper accordingly. Please see our updated PDF.
> > >
> > > Let us know if there's any further concerns. We would be happy to adopt your suggestions.

---

> > > > ### Comment · Reviewer_u9zJ · 2023-08-22
> > > >
> > > > I have raised the score to BA because the authors have rephrased the semantics part and incorporated new references.

---

### Official Review · Reviewer_2WCT · 2023-07-24
**A novel and simple error metric for evaluating camera trajectories in the wild.**

**Rating:** 6
**Confidence:** 4

**Strengths:**

- Simplicity and Practicality: The proposed Object Reprojection Metric (ORE) stands out as a straightforward and user-friendly evaluation protocol, particularly in scenarios where 3D ground truth data is unavailable. Its simplicity makes it a valuable tool for assessing camera trajectories in a wide range of applications.
- Comprehensive Dataset (EgoStatic): The EgoStatic dataset is extensive, providing ample data for training and evaluating large-scale deep learning models.
- Comparison with Established Metrics: The paper includes a thorough comparison with established metrics like ATE and RPE, enhancing the credibility of ORE.
- Readability: The paper is easy to read and accessible.
- Comprehensive Experiments: The comprehensive experiments on ScanNet and EgoStatic, along with multiple baselines, establish the correlation of ORE with standard metrics.

**Additional Feedback:**

N/A. Please see above.

**Clarity:**

Yes, the paper is mostly clear to read. However, there are some minor confusions that need to be addressed:

- [Line 225] Please refer to the second comment in the "Opportunities for Improvement" section regarding the confusion around Line 225.
- [Line 258] In some instances, there is confusion about the presentation of numbers, such as "0.097 ORE (+0.005)" on Line 258. The meaning of the number within the bracket is unclear. If you are color-coding the numbers in tables, it would be helpful if you do the same on in-line texts. This could help improve clarity and consistency.
- [Line 255] There is confusion when discussing the performance of different versions of COLMAP, such as Default, Tuned [62, 76], and Mask [89], and their associated ORE values. Clarity is needed to understand which version performs better, as statements seem contradictory in some cases. The next statement states that Default achieves 0.097 ORE (+0.005). Does it mean default is better? If so, the next statement "This suggests the practice introduced by [76] (which is a method of Tuned) is indeed improving camera trajectory qualities." implies that Tuned is better. Moreover, the next statements in Line 260 again introduces the confusion regarding performance of Mask. Revisiting these statements could help reduce confusion for readers.

By addressing these points, the paper's clarity can be further improved, ensuring a better understanding of the presented results and findings.

**Correctness:**

The claims made in the submission appear to be correct. The dataset construction from Ego4D and the creation of EgoStatic as a subset seem appropriate. The benchmark, evaluation methods, and experiment design are suitable for the purpose. Any areas of uncertainty or questions raised have been highlighted in the "Opportunities for Improvement" section mentioned above.

**Documentation:**

The authors have sourced the EgoStatic dataset from the existing Ego4D dataset, and as such, the majority of responsibility regarding documentation, availability, and maintenance lies with Ego4D. The authors have provided a code repository (in supplementary) responsible for extracting EgoStatic from Ego4D, and it is crucial for them to publish and maintain this code repository online for future reference and reproducibility. Reviewers and the research community will greatly benefit from having access to the code repository to ensure transparency and replicability of the results.

**Ethics:**

No.

**Limitations:**

The authors have adequately addressed the limitations of their work in Section 5 of the paper.

**Opportunities For Improvement:**

- The paper does not explicitly clarify whether all static objects within the EgoStatic benchmark are chosen for evaluation. An ablation study on the number of objects used (increasing or decreasing) could provide insights into the significance of individual objects on the evaluation results.

- Line 225-226 introduces confusion regarding the importance of COLMAP for depth estimation. The statement in line 224 suggests that depth estimation for a worse-performing method would be less accurate. However, Table 2 indicates that COLMAP performs worse than DROID-SLAM. Further clarification on the relationship between the methods and their depth estimation accuracy would be helpful.

- The assertion that fine-tuning did not improve EgoStatic warrants further investigation. Conducting a study on an ego-centric dataset similar to Ego4D and evaluating the impact of fine-tuning could shed light on potential distribution differences and their effects on performance.

- The caption of Table 5 raises an interesting question about COLMAP's superior performance over data-driven methods in EgoStatic. Understanding the factors contributing to this behavior and comparing it to other ego-centric datasets could provide valuable insights. The authors' perspective on this matter would be beneficial.

**Relation To Prior Work:**

Yes.

**Summary And Contributions:**

Summary: The paper introduces a novel evaluation protocol called Object Reprojection Metric (ORE) to assess the accuracy of estimated camera trajectories. In scenarios where obtaining ground truth 3D information is challenging, ORE leverages lightweight object tracklet annotations, eliminating the need for precise 3D data. By reprojecting object tracklets into 2D, ORE effectively evaluates camera trajectories. The proposed error metric is validated through comparisons with established standard metrics. Utilizing ORE, the authors curate the EgoStatic dataset, sourced from Ego4D-EgoTracks, which serves as a large-scale, diverse benchmark for evaluating camera trajectories in various environments.

Contributions:
- Introduction of the novel evaluation protocol ORE, providing an effective and accessible error metric for assessing camera trajectories without relying on precise 3D ground truth information.
- Benchmarking and validation of the proposed ORE error metric against established standard metrics, demonstrating its accuracy and reliability.
- Creation of the EgoStatic benchmark dataset using ORE, which offers a challenging and diverse dataset for evaluating camera trajectories in real-world scenarios.

---

> ### Author Response · Authors · 2023-08-15
> **Response to Reviewer 2WCT**
>
> > Q1:The paper does not explicitly clarify whether all static objects within the EgoStatic benchmark are chosen for evaluation. An ablation study on the number of objects used (increasing or decreasing) could provide insights into the significance of individual objects on the evaluation results.
>
> A1: For the videos within the categories we evaluated, all static objects are used.  We will clarify in the updated version. In the evaluation protocol, we use all static objects annotated within the EgoStatic benchmark for ORE evaluation. On average, each video contains 3.67 static object tracklets.
>
> We would also like to refer the reviewer to our study in Table 4, where we ablate on the size of bounding boxes. This sheds light on what type of objects are preferred in the evaluation.
>
> In addition, we provide a further ablation study about the number of objects to show the positive correlation between the increase of the number of tracklets and the accuracy of the benchmark.
> Specifically, we gradually increase the number of tracklet selected in the ScanNet dataset from 1 to 5 and observe that the higher number of tracklets we selected, the better the correlation between ORE and ATE will be.
> #### Table: Ablation on box number for ORE vs. ATE_rot (7 models)
> | Box Number    |     1 |     3 |     5 |
> |---------------|:------:|:------:|:------:|
> | Spearman      | 0.771 | 0.836 | 0.855 |
> | Kendall's tau | 0.643 | 0.700 | 0.729 |
>
> > Q2: Line 225-226 introduces confusion regarding the importance of COLMAP for depth estimation. The statement in line 224 suggests that depth estimation for a worse-performing method would be less accurate. However, Table 2 indicates that COLMAP performs worse than DROID-SLAM. Further clarification on the relationship between the methods and their depth estimation accuracy would be helpful.
>
> A2: We would clarify this further in the paper.
> In the ScanNet dataset, COLMAP performs worse than DROID-SLAM as shown in Table 2. This gap in performance is also reflected in Figure 3 and described in Line225-6, since COLMAP’s optimized depth is further from ground truth compared with DROID-SLAM (COLMAP has fewer sequences with correct optimal depth ). These two observations agree with each other to validate our assumption in “depth value found for a worse-performing method will be less accurate”.
>
> > Q3: The assertion that fine-tuning did not improve EgoStatic warrants further investigation. Conducting a study on an ego-centric dataset similar to Ego4D and evaluating the impact of fine-tuning could shed light on potential distribution differences and their effects on performance.
>
> A3: We thank the reviewer for providing this suggestion. In Table 6 of “Domain adaptation of MonoDepth2”, we show that fine-tuning MonoDepth2 (originally trained on driving data KITTI) on more in-domain dataset like NYUv2 (indoor scene) improves the performance of Mondepth2 on both ScanNet and EgoStatic.
>
> In addition, we adopt the suggestion from the reviewer to further study the effect of fine tuning on egocentric video. We find that naive fine-tuning MonoDepth2 on a single/a small subset of video in Ego4D would degenerate the model performance due to irregular motion existing in Ego4D compared to scan-like motion in NYUv2 or linear motion in KITTI. Thus we adopt multiple tricks including sampling frames with appropriate motion, freezing pose or depth modules during training etc., which led to improvement of ORE in several videos we reported below. However, the performance of the finetuned model drops when testing on other videos from EgoStatic. This highlights a potential challenge in generalization.
>
> We believe that large scale experiments is a potential direction; meanwhile, we consider how to properly fine-tune MonoDepth2 on Ego4D out of the scope for this paper and deserve its own research but we will be happy to provide additional experiments or studies if the reviewer deem these experiments critical for the paper’s completeness or is willing to suggest the settings we are missing.
>
> | Training data         | Trained on kitti | Trained on Ego4D |
> |-----------------------|:------------------:|:------------------:|
> | ORE tested on video 1 |            0.441 |            0.154 |
> | ORE tested on video 2 |            0.532 |            0.224 |
>
> We will be happy to provide more experiments or studies if the reviewer could suggest the settings we are missing.

---

> > ### Author Response · Authors · 2023-08-15
> > **Response to Reviewer 2WCT (cont.)**
> >
> > > Q4: The caption of Table 5 raises an interesting question about COLMAP's superior performance over data-driven methods in EgoStatic. Understanding the factors contributing to this behavior and comparing it to other ego-centric datasets could provide valuable insights. The authors' perspective on this matter would be beneficial.
> >
> > A4: We believe there are a few difficulties with the learned methods compared to COLMAP. First, learned methods generalize worse and tend to overfit to the data they are trained on. For example, as stated in the answer to previous question, finetuning MonoDepth2 on one video will worsen its performance on others. In contrast, COLMAP is learning-free and geometric-based, and is less prone to overfit to specific data. In addition, COLMAP has many carefully designed mechanisms to handle outliers or drifting, which makes it more robust to motion blur in EgoStatic dataset. Adopting this mechanism to learning-based methods can be an interesting future direction. Finally, the advantage of COLMAP has also been observed qualitatively in previous dataset, such as EPIC-Kitchen. For example, NeuralDiff [94] chooses COLMAP over other methods to obtain camera trajectories for neural rendering.
> >
> > [94] Tschernezki, Vadim, Diane Larlus, and Andrea Vedaldi. "NeuralDiff: Segmenting 3D objects that move in egocentric videos." 2021 International Conference on 3D Vision (3DV). IEEE, 2021.

---

### Official Review · Reviewer_i2aG · 2023-07-25
**Review: Object Reprojection Error (ORE)...**

**Rating:** 7
**Confidence:** 4
**Correctness:** I believe the claims to be correct.
**Clarity:** Aside from points noted above, the pa…

**Strengths:**

1. Clever re-purposing of existing large dataset + all the work needed to find good static object tracks makes a useful new benchmark dataset.
2. Good demonstration that the benchmark evaluation is useful by comparing to existing benchmarks, showing there is room for improvement, and demonstrating that the error function can be used to fine-tune other models.


**Additional Feedback:**

NA

**Documentation:**

yes.

**Ethics:**

no ethics concerns.

**Limitations:**

2. from the paper: “all methods perform poorly on Social and Handyman. Social  often involves many moving people, increasing the difficulty to identify good static correspondences across frames; Handyman involves large movement in an indoor environment, often with motion blur."

This could be because the dataset if challenging.  Or it could be because the ORE benchmarking approach fails if you can’t find suitable static features

**Opportunities For Improvement:**

1. I found Table 2 confusing for a long time.

I think each of the three rows have different units and are not really compariable?

ORE is measured in “pixels” (normalized by the image size), ATE is measured in 3D coordinate system units, and RPE is measured in something related to angles (radians?)

 Additionally, ORE gives a score of 0 for any reprojection inside the object bounding box while the other approaches have a specific target location.  So, sensible that section 4.2.2 finds the methods match more closely when the objects are small --- but confusing to put them into the same table in ways that encourage comparisons between the numbers.

2. I would love to have more discussion of the comparison to geometric reprojection error --- having a user search for "small" static objects is relatively similar to having a user identify static "easy to track" points, or static "planar surfaces" --- and then you would get rid of all the awkwardness of having zero error no matter where you project inside a bounding box.

**Relation To Prior Work:**

yes.

**Summary And Contributions:**

Summary: This paper considers the problem of SLAM/SfM and offers a new metric to judge the accuracy of a set of computed camera-poses.  Unlike classical methods that require comparison to ground truth, the suggestions is to hand annotate static objects and compute the bounding box of those objects throughout the video.  Then, if the set of camera poses is good, the depth of each object can be estimates to give a 3d position that can be projected to every other frame, with error accumulating for frames where it reprojects outside the object bounding box.

A dataset is created of >5000 videos with over 22,000 hand-annotated “static object tracks”.  They demonstrate that this metric correlates well with current metrics that may require 3D groundtruth which is not often availabe.

---

> ### Comment · Reviewer_u9zJ · 2023-08-06
> **Geometric Reprojection**
>
> I think i2aG's argument on geometric reprojection is very interesting. Selecting small objects turns to selecting static salient points in the extreme case. Now I wonder how ORE compares to GRE too.

---

> ### Author Response · Authors · 2023-08-15
> **Response to Reviewer i2aG**
>
> > Q1: I found Table 2 confusing for a long time.
>
> A1: We thank the reviewer for carefully reviewing the details. Your understanding is correct that the units for the three metrics are different. We are thinking about adding the unit in the column header as well as adding lines between the rows to distinguish them. The goal of this table is to show that the rank order of the methods is largely consistent across the metrics (rows). We will reflect this in the caption of the table. Please let us know if there are better ways to avoid confusions and we will gladly adopt your suggestions.
>
>
>
> > Q2: I would love to have more discussion of the comparison to geometric reprojection error --- having a user search for "small" static objects is relatively similar to having a user identify static "easy to track" points, or static "planar surfaces" --- and then you would get rid of all the awkwardness of having zero error no matter where you project inside a bounding box.
>
> A2: We thank the reviewer for your insightful thoughts; indeed, “easy to track” points are a special case of ORE when the “object” chosen is small enough and can be treated as a point.
>
> First, we would like to refer reviewers to our study in Table 4 regarding smaller bounding box size.  Smaller size improves rank correlation between ORE and classical metrics (ATE/RPE). We consider geometric reprojection error (GRE) as a special case where the bounding box size reaches zero (collapsed into an “easy-to-track” point).
> We implemented GRE metrics following the reviewer's description and computed the ranking correlation between GRE and classical metrics. We found the correlation to improve (albeit marginally) compared to small-sized bounding boxes (compared with the last column in Table 4). In other words, Table 4 + added GRE experiment shows that ORE effectively converges to GRE in the limit of size-zero boxes, as one would hope.
>
> #### Table: Ranking correlation between GRE and standard metrics
> |              |   Spearman | Kendall’s $\tau$|
> |----------|:-------------:|:------:|
> |ATE_trans |   0.720        |  0.610 |
> |ATE_rot     |   0.800        |  0.662|
>
>
> Meanwhile, we remark several practical reasons for using static object tracklets over point tracklet annotation.
>
> 1. Points can lose track more easily than objects. This is especially common in egocentric videos with occlusion, large viewpoint change and motion blur. For example, when the camera turns to the back of an object, the point initialized at the front side will not be visible for annotation. In contrast, the object bounding box can be annotated accurately. Therefore, object tracklets have higher coverage than points throughout the video.
> 2. Object annotations are more accessible than point-tracking annotations. There is a large collection of existing object tracking annotations, such as Ego4D, EPIC-VISOR, FPV, etc. The advantage of ORE is that we can evaluate videos from these datasets with minimal additional annotation (e.g. static or moving). On the other hand, existing point-tracking dataset is less common. Recent TAP-Vid [93] dataset focuses mainly on third-person videos, with very little or no camera movement.
> 3. Object annotations require less effort. As pointed out in TAP-Vid paper, “,annotating point tracks in real videos is extremely time consuming, since both objects and cameras tend to move in complex, non-linear ways”.
>
> [93] Doersch, Carl, et al. "Tap-vid: A benchmark for tracking any point in a video." Advances in Neural Information Processing Systems 35 (2022): 13610-13626.
>
> > Q3: This could be because the dataset is challenging. Or it could be because the ORE benchmarking approach fails if you can’t find suitable static features
>
> A3: We thank the reviewer for pointing out the alternative angle for explaining the failures. Quantitatively, on average, these two categories have 4.14 static object tracklets, in fact higher than 3.67 static object tracklets for EgoStatic as a whole. Since both the object tracklets and the static / dynamic attributes are labeled by humans, we believe that these videos do have good static object tracklets available but are more difficult than others.
>
> We also manually confirmed that these videos do indeed have a good pool of candidates for static object tracklets. For example, videos in Social include scenes such as people hosting a party in a kitchen, where objects such as kettles, containers are static throughout the video.  We believe the difficulty arises at least in part from the scene being occupied largely by dynamic components such as humans. These dynamic pixels present serious challenges to existing approaches.

---

### Official Review · Reviewer_U9Jy · 2023-07-28
**Interesting addition to the literature**

**Rating:** 8
**Confidence:** 4
**Correctness:** Seems correct to me.
**Clarity:** Very clear writing and presentation.

**Strengths:**

Very clear, well written paper, with a useful, smart contribution for this field that makes it possible to better evaluate camera pose estimation methods on videos in the wild.

Good ablations.

**Additional Feedback:**

Good work!

**Documentation:**

All good.

**Ethics:**

All good.

**Limitations:**

Limitations discussed in the paper.

**Opportunities For Improvement:**

I didn't find that many issues. If one really wanted to improve the paper, one thing that could be done would be to compare the effort and end result of manually annotating pose vs boxes on some sequences from e.g. Scannet, for which there is ground truth and repeat the rank correlation study to see the trade-off between annotation time and evaluation quality.

**Relation To Prior Work:**

Discussed.

**Summary And Contributions:**

This is a very interesting paper for evaluation of camera pose estimation. It proposes to use static box tracks as only required annotation for evaluating camera pose, which can be in principle be more easily obtained for in the wild scenes or existing footage on the web.

The paper contributes a benchmark based on Ego4D, an evaluation method, called ORE, which typically requires an optimization loop where it solves for depth for a point inside each box. It also shows that ORE correlates well with other existing metrics on a dataset for which there is annotations (ScanNet) in terms of the way it ranks different pose estimation methods.

---

> ### Author Response · Authors · 2023-08-15
> **Response to Reviewer U9Jy**
>
> We thank the reviewer for the positive feedback and comments.
>
> We thank the reviewer for suggesting a comparison between obtaining static object tracklets versus annotating poses. We would like to remark that camera poses are hard if not impossible to annotate accurately. Existing dataset either uses synthetic data, such as TartanAir [83], or uses SLAM sensors including IMU, depth etc. to acquire poses, such as ScanNet [16] and KITTI [27] . This makes camera pose data nearly impossible to collect on existing real-world video datasets without such groundtruth.

---

> > ### Comment · Reviewer_U9Jy · 2023-08-29
> > **Don't think it's impossible, it's been done in some form or other**
> >
> > Datasets like https://cvgl.stanford.edu/projects/pascal3d.html and https://www2.eecs.berkeley.edu/Research/Projects/CS/vision/human/poselets_iccv09.pdf annotated 3d pose in static frames, so it would not be a huge stretch to annotate one sequence just to get how much worse that would be.

---

> > > ### Author Response · Authors · 2023-08-29
> > > **Response to Reviewer U9Jy**
> > >
> > > We thank the reviewer again for suggesting annotating 3D poses. We agree that object or third-person view of human body pose is possible to annotate, as pointed out by the references. However, we would like to point out that camera pose is very difficult to annotate, and prior works rely on SLAM sensors including IMU, depth etc. to collect camera pose trajectory.
> > >
> > > On the other hand, the reviewer’s comment points out to an interesting future direction. It is possible to annotate 3D bounding boxes on static objects, and then recompute groundtruth camera pose from the 3D bounding boxes: rotation can be computed directly and translation may require groundtruth depth information. We believe this can be an interesting alternative way but it can be a new benchmark method by itself.

---

### Author Response · Authors · 2023-08-15
**General response to reviewers**

We appreciate all reviewers’ review and positive feedback. We thank the reviewers for finding our work clear and well-written, for acknowledging the ORE evaluation protocol as a simple, useful, smart, and novel way to benchmark camera pose estimation methods and for finding our experiments extensive and well-supported with ablations.

We here provide initial responses for questions raised in the reviews; if reviewers find the information in the responses useful, we plan to revise the paper to reflect the discussions upon feedback received.

---

### Author Response · Authors · 2023-08-22
**First Revision of Paper**

We thank the reviewer for all useful comments and suggestions. We hereby provide a first revision of the paper. The new additions and changes are highlighted in red and the summarization of the changes are listed below:
1. We modified the introduction to reflect concerns on spatial memory motivation.
2. We added proper discussion and comparison with works on visual re-localizations in Sec 2.1
3. We revised Table 2 to reflect the confusions on each metrics’ units.
4. We added additional discussion regarding comparison with (Geometric) Reprojection Error in Sec 4.2.2.
5. We added Table 6 and rephrased Sec 4.3.2 to improve readability on Design choices in COLMAP.

After discussion with reviewers, we plan to add all other ablations to the supplementary material and polish the paper again to adopt additional suggestions. Please let us know if there’s any further concerns.

---

### Author Response · Authors · 2023-08-25
**General response to reviewers**

We sincerely thank the reviewers again for their helpful suggestions and insightful feedback.

We would like to check if there’s any further concerns and additional comments for the paper we could potentially address before the end of the rebuttal period. We are always happy to provide further discussions and ablations.

---

### Decision · Program_Chairs · 2023-09-22

**Decision:**

Accept (Poster)

**Comment:**

This paper proposes a new metric for camera pose estimation, especially for ego-centric views. Reviewers find the paper well written and the method and experiments solid. The authors have actively engaged in discussion and revised the paper accordingly. In conclusion, this paper can be accepted.